# APPROXIMATION AND NON-PARAMETRIC ESTIMATION OF FUNCTIONS OVER HIGH-DIMENSIONAL SPHERES VIA DEEP RELU NETWORKS

**Namjoon Suh, Tian-Yi Zhou, Xiaoming Huo**
H.Milton Stewart School of Industrial and Systems Engineering
Georgia Institute of Technology, Atlanta, GA, USA
{namjsuh,tzhou306,huo}@gatech.edu

## ABSTRACT

We develop a new approximation and statistical estimation analysis of deep feed-forward neural networks (FNNs) with the Rectified Linear Unit (ReLU) activation. The functions of interests for the approximation and estimation are assumed to be from Sobolev spaces defined over the $d$-dimensional unit sphere with smoothness index $r > 0$. In the regime where $r$ is of the constant order (i.e., $r = \mathcal{O}(1)$), it is shown that at most $d^d$ active parameters are required for getting $d^{-C}$ approximation rate for some constant $C > 0$. In the regime where the index $r$ grows in the order of $d$ (i.e., $r = \mathcal{O}(d)$) asymptotically, we prove the approximation error decays in the rate $d^{-d^\beta}$ with $0 < \beta < 1$ up to some constant factor independent of $d$. The required number of active parameters in the networks for the approximation increases polynomially in $d$ as $d \to \infty$. It is also shown that bound on the excess risk has a $d^d$ factor, when $r = \mathcal{O}(1)$, whereas it has $d^{\mathcal{O}(1)}$ factor, when $r = \mathcal{O}(d)$. We emphasize our findings by making comparisons to the results on the approximation and estimation errors of deep ReLU FNN when functions are from Sobolev spaces defined over $d$-dimensional cube. In this case, we show that with the current state-of-the-art result, $d^d$ factor remain both in the approximation and estimation errors, regardless of the order of $r$.

## 1 INTRODUCTION

Neural networks have demonstrated tremendous success in the tasks of image classification (Krizhevsky et al., 2012; Long et al., 2015), pattern recognition (Silver et al., 2016), natural language processing (Graves et al., 2013; Bahdanau et al., 2015; Young et al., 2018), etc. The datasets used in these real world applications frequently lie in high-dimensional spaces (Wainwright, 2019). In this paper, we try to understand the fundamental limits of neural networks in the high-dimensional regime through the lens of its approximation power and its generalization error.

Both approximation power and generalization error of neural network can be analyzed through specifying the target function's property such as its smoothness index $r > 0$ and its input space $\mathcal{X}$. In particular, deep feed-forward neural networks (FNNs) with Rectified Linear Units (ReLU) have been extensively studied when they are used for approximating and estimating functions from general function class such as Sobolev class defined on $d$-dimensional cube (i.e., $\mathcal{X} := \mathcal{C}^d$), denoted as $W_p^r(\mathcal{C}^d)$ for $1 \leq p \leq \infty$. However, in practice, signals on a spherical surface (i.e., $\mathcal{X} := \mathcal{S}^{d-1} = \{\mathbf{x} \in \mathbb{R}^d : \|\mathbf{x}\|_2 = 1\}$) rather than on Euclidean spaces often arise in various fields, such as astrophysics (Starck et al., 2006; Wiaux et al., 2005), computer vision (Brechbühler et al., 1995), and medical imaging (Yu et al., 2007).

Motivated by this, we focus our attention on the cases where deep ReLU FNNs are used for function approximators and estimators, when functions are assumed to be from the Sobolev spaces defined over $\mathcal{S}^{d-1}$; that is $f \in W_\infty^r(\mathcal{S}^{d-1})$. Under this setting, our analysis focuses on how the input dimension $d$ explicitly affects the approximation and estimation rates of $f \in W_\infty^r(\mathcal{S}^{d-1})$. At the same time, we show how the scalability of deep ReLU FNNs grows in the high-dimensional regime. Here, the scalability is mainly measured through the three metrics: (1) *the width denoted as* $\mathcal{W}$,

|  | Theorem 4.3 | | Theorem 4.4 |
|---|---|---|---|
| Function class | $W_\infty^r(\mathcal{S}^{d-1})$ | | $W_\infty^r([0,1]^d)$ |
| Smoothness $r$ | $\mathcal{O}(d)$ | $\mathcal{O}(1)$ | $\forall r > 0$ |
| Upper-bound on $\mathcal{N}$ | $\mathcal{O}(nd)$ | $\mathcal{O}(nd)$ | $\tilde{\mathcal{O}}((d+r)^d)$ |
| Estimation error rate | $\tilde{\mathcal{O}}\big(d^C \cdot n^{-\frac{4r}{4r+3d}}\big)$ | $\tilde{\mathcal{O}}\big(\big(\frac{6}{\pi e}\big)^{\frac{d}{2}} d^d \cdot n^{-\frac{4r}{4r+3d}}\big)$ | $\tilde{\mathcal{O}}\big((d+r)^d \cdot n^{-\frac{2r}{2r+d}}\big)$ |

Table 1: Here, $C > 0$ is an universal constant. Notation $\tilde{\mathcal{O}}(\cdot)$ hide the logarithmic factor in $n$. Note that the upper-bounds for $\mathcal{N}$ in Theorem 4.3 (i.e., $\mathcal{N} = \mathcal{O}(Md)$) are from Theorem 3.1 with choices $M = \lceil n^{\frac{3d}{3d+4r}} \rceil$.

(2) *the depth, denoted as $L$,* and (3) *the number of active parameters, denoted as $\mathcal{N}$* of the network, (Anthony & Bartlett, 1999). It should be emphasized that we find there exists an interaction with smoothness index $r > 0$ and dimension $d$, whereas we cannot find one for the case when $f \in W_\infty^r(\mathcal{C}^d)$. We further summarize our detailed findings in the following Subsection.

## 1.1 PAPER ROAD MAP AND CONTRIBUTIONS

In Theorem 3.1, we provide an approximation bound of deep ReLU FNN (i.e., $\tilde{f}$) for approximating the target functions in Sobolev spaces defined over sphere (i.e., $f \in W_\infty^r(\mathcal{S}^{d-1})$). Notably, in the bound, we track the explicit dependence on data dimension $d$ allowing it tends to infinity. This tracking enables how the three components of network architecture, **width** ($\mathcal{W}$), **depth** ($L$), and **the number of active parameters** ($\mathcal{N}$), should change as $d$ increases, for obtaining the good approximation error rate.

As a Corollary of Theorem 3.1, we show how the order of function smoothness $r$ can have the effect on the scale of network in terms of $d$. Specifically, when the function smoothness $r = \mathcal{O}(1)$, we show that the constructed network, $\tilde{f}$, requires $\mathcal{W} = \mathcal{O}(d^d)$, $L = \mathcal{O}(d^\gamma \log_2 d)$ for $0 < \gamma < 1$, and $\mathcal{N} = \mathcal{O}(d^{d+1})$ for obtaining $d^{-\mathcal{O}(1)}$ approximation error up to some constant factors independent with $d$. Furthermore, when $r = \mathcal{O}(d)$, we show that only $\mathcal{W} = \mathcal{O}(d^\alpha)$, $L = \mathcal{O}(d^\gamma \log_2 d)$, and at most $\mathcal{N} = \mathcal{O}(d^2)$ are required for obtaining the sharp approximation rate $\mathcal{O}(d^{-d^\beta})$ for $0 < \alpha, \beta < 1$. See Corollary 3.3 for the detailed statement of the result.

Our result implies that for approximating $f \in W_\infty^r(\mathcal{S}^{d-1})$, the larger the smoothness index $r$ is, the narrower the width of the network should be enough, while the depth of the network can be fixed. Moreover, when $r$ is in the same order as $d$, the network can avoid the **curse of dimensionality** requiring only $\mathcal{O}(d^2)$ number of active parameters. It is interesting to note that the function smoothness index can affect the design of the network, specifically on width, while it has little effect on the design of depth. Admittedly, the condition $r = \mathcal{O}(d)$ is restrictive in a sense that it makes the function space $W_\infty^r(\mathcal{S}^{d-1})$ small. Nonetheless, it contains some interesting examples: that is, reproducing kernel Hilbert spaces (RKHS) generated by $C^\infty$ kernels such as Gaussian kernels.

Additionally, to the best of our knowledge, this finding is not observed in the current approximation theory of neural network literature when $f \in W_\infty^r(\mathcal{C}^d)$ where $\mathcal{C}^d$ denotes some $d$-dimensional cubes, and $\tilde{f}$ is a deep ReLU FNN. Out of the long list of literature to be introduced shortly, we choose the result from Schmidt-Hieber (2020) for the comparison as it also has the explicit dependence on $d$ in their approximation bound. From their result, it can be seen that the **curse** cannot be avoided, even when $r = \mathcal{O}(d)$. The width of their constructed network is lower bounded by $\Omega(r^d \vee e^d)$ and the number of active parameters is upper-bounded by $\mathcal{O}((r+d)^d)$. [1] Note that the bounds on both components grow exponentially in $d$ as $r$ increases. See Subsection 3.1 for detailed comparisons.

We further make the comparisons between estimating functions $f \in W_\infty^r(\mathcal{S}^{d-1})$ (Theorems 4.3) versus $f \in W_\infty^r(\mathcal{C}^d)$ (Theorems 4.4) via deep ReLU FNNs under the non-parametric regression framework. Given $n$ noisy samples, the two Theorems suggest the specific orders of $\mathcal{W}$, $L$ and $\mathcal{N}$ in terms of $n$, $d$ and $r$, for which they give the tightest bound on excess risk of respective function estimator from Proposition 4.2. When $r = \mathcal{O}(1)$, it is shown that the excess risk upper-bounds of

---

[1] Interested readers can find the intuitive technical reason for having the exponential dependence in $d$ on width $\mathcal{W}$ and active parameters $\mathcal{N}$ in the Appendix A.

both function estimators have $d^d$ in the constant factors. In contrast, when $r = \mathcal{O}(d)$, estimating functions $f \in W_\infty^r(\mathcal{S}^{d-1})$ has at most $d^{\mathcal{O}(1)}$ factor in the bound, whereas the bound for function estimator of $f \in W_\infty^r(\mathcal{C}^d)$ has $d^d$. See Table 1 and Subsection 4.2 for detailed comparisons.

## 1.2 RELATED WORKS

In this Subsection, to aid readers have a more clear understanding on the contributions of our paper, we provide the list of relevant works with comparisons of how these works are different from ours.

***Approximation of*** $f \in W_\infty^r(\mathcal{S}^{d-1})$ ***via deep CNN.*** For the approximation theory of $f \in W_\infty^r(\mathcal{S}^{d-1})$, we must refer readers Fang et al. (2020) and Feng et al. (2021). But in their works, the convolutional neural network (CNN) is used for the function approximator under fixed $d$ setting.

***Approximation of*** $f \in W_\infty^r(\mathcal{C}^d)$ ***via deep ReLU FNN.*** Approximation theory of deep ReLU FNN for functions $f \in W_\infty^r(\mathcal{C}^d)$ has a lengthy history in the literature. Representatively, Mhaskar (1996) showed that $f$ can be approximated uniformly within $\varepsilon$-approximation accuracy with a 1-layer neural network of $\mathcal{O}(\varepsilon^{-d/r})$ neurons and an infinitely differentiable activation function. Later, for deep ReLU networks, Yarotsky (2017) showed that the number of active parameters $(\mathcal{N})$ in networks is bounded by $\mathcal{O}(\varepsilon^{-d/r} \log(\frac{1}{\varepsilon}))$, and the depth has the order $\mathcal{O}(\log(\frac{1}{\varepsilon}))$. He further proved that $\mathcal{N}$ is lower-bounded by the order $\mathcal{O}(\varepsilon^{-d/r})$, which is backed up by the result in DeVore et al. (1989). For $f \in W_p^r(\mathcal{C}^d)$ with $1 \leq p \leq \infty$, Petersen & Voigtlaender (2018) showed that there exists a deep ReLU network with bounded and quantized weight parameters, with $\mathcal{O}(\varepsilon^{-d/r})$ network size, and with $\varepsilon$-independent depth for achieving the $\varepsilon$-accuracy in the $L_p$ norm. For approximating functions $f \in W_\infty^r(\mathcal{C}^d)$, Schmidt-Hieber (2020) proved that a network of size $\mathcal{O}(\varepsilon^{-d/r})$ with bounded weight parameters achieves $\varepsilon$-approximation error in the $L_\infty$ norm.

***Function spaces with special structures.*** The result of Yarotsky (2017) implies that deep ReLU net cannot escape the curse of dimensionality for approximating $f \in W_\infty^r(\mathcal{C}^d)$. Many papers have demonstrated that the effects of dimension can be either avoided or lessened by considering function spaces different from Sobolev spaces, but defined over $\mathcal{C}^d$. Just to name a few, Mhaskar et al. (2016) studied that a function with a compositional structure with regularity $r$ can be approximated by neural network with $\mathcal{O}(\varepsilon^{-2/r})$ neurons within $\varepsilon$ accuracy. Suzuki (2018) proved the deep ReLU network with $\mathcal{O}(\varepsilon^{-1/r})$ neurons can avoid the curse for approximating functions in mixed smooth Besov spaces. Chen et al. (2019) showed the network size scales as $\mathcal{O}(\varepsilon^{-D/r})$ for approximating $C^r$ functions, when they are defined on a Riemannian manifold isometrically embedded in $\mathbb{R}^d$ with manifold dimension $D$ with $D \ll d$. Montanelli & Du (2019) and Blanchard & Bennouna (2022) showed respectively the deep and shallow ReLU network break the curse for Korobov spaces.

***Estimation rates of excess risk under non-parametric framework.*** Many researchers also have tried to tackle how the neural networks avoid the curse by considering specially designed function spaces under the non-parametric regression framework. We only provide an incomplete list of them. Such structures include additive ridge functions (Fang & Cheng, 2023), composite function spaces with hierarchical structures (Schmidt-Hieber, 2020; Han et al., 2022), mixed-Besov spaces (Suzuki, 2018), Hölder spaces defined over a lower-dimensional manifold embedded in $\mathbb{R}^d$ (Chen et al., 2022). They all showed the function estimators with neural network architectures can lessen the curse by showing the excess risks of the estimators are bounded by $\mathcal{O}(n^{-2r/(2r+D')})$, where $n$ denotes the size of a noisy dataset, and $D' \ll d$ is an intrinsic dimension uniquely determined through the characteristics of function spaces, when they are compared with the minimax risk $\mathcal{O}(n^{-2r/(2r+d)})$ (Donoho & Johnstone, 1998) for $f \in W_\infty^r(\mathcal{C}^d)$.

***Comparisons.*** The aforementioned works mainly focused on the approximation and estimation of functions defined on $\mathcal{C}^d$, not $\mathcal{S}^{d-1}$, for the fixed $d$. Moreover, the introduced papers on approximation theory, except the work of Schmidt-Hieber (2020), hide the dependence on $d$ in the Big-$\mathcal{O}$ notation of $\mathcal{N}$ in $\varepsilon$-accuracy, even for papers where they consider the function spaces with special structures. Thus, it is not clear how the $d$ affects the approximation bound and the scale of the provided network architecture. Introduced papers on estimation rate for excess risk also follow the same philosophy with papers on approximation theory, as they work on the fixed $d$ setting. In contrast, we work on the $\mathcal{S}^{d-1}$ input space, track the explicit dependence on $d$ in the error bound, and describe

how $d$ affects the scale of deep ReLU FNN as $d \to \infty$ with its interactions with function smoothness $r > 0$. Our paper focuses on tracking the dependence on $d$ in the constant factor hidden in the Big-$\mathcal{O}$ notations both in approximation and estimation error rates, rather than paying attentions to reducing the exponential dependence of $d$ with base $\varepsilon$ in $\mathcal{N}$ or with base $n$ in excess risk bound.

## 2 PRELIMINARY DEFINITIONS

In this Section, we provide the mathematical definitions of deep ReLU FNN and Sobolev function spaces on unit sphere.

### 2.1 DEFINITION OF DEEP ReLU NETWORK

For defining the deep ReLU network mathematically, we adopt the notation used in Schmidt-Hieber (2020). For $\mathbf{v} = (\mathbf{v_1}, \ldots, \mathbf{v_r}) \in \mathbb{R}^r$, let $\sigma_{\mathbf{v}} : \mathbb{R}^r \to \mathbb{R}^r$ be the shifted ReLU (Rectified Linear Units) activation function as $\sigma_{\mathbf{v}}((y_1, \ldots, y_r)^\top) := \sigma((y_1 - \mathbf{v}_1, \ldots, y_r - \mathbf{v}_r)^\top)$, where $\sigma(x) = max(x, 0)$.

With this notation, the network architecture $(L, \mathbf{p})$ consists of a positive integer $L$, called *the number of hidden layers*, and *a width vector* $\mathbf{p} := (\mathbf{p}_0, \ldots, \mathbf{p}_{L+1}) \in \mathbb{N}^{L+2}$. A deep ReLU network with architecture $(L, \mathbf{p})$ considered in this work is then any function of the form

$$\tilde{f} : \mathcal{S}^{d-1} \to \mathbb{R}, \quad \mathbf{x} \to f(\mathbf{x}) = W_L \sigma_{\mathbf{v_L}} W_{L-1} \sigma_{\mathbf{v_{L-1}}} \ldots \sigma_{\mathbf{v_1}} W_1 \mathbf{x}, \tag{1}$$

where $\mathbf{W}_i \in \mathbb{R}^{p_{i+1} \times p_i}$ is a weight matrix with $\mathbf{p}_0 = d$, $\mathbf{p}_{L+1} = 1$ and $\mathbf{v}_i \in \mathbb{R}^{p_i}$ is a shift vector. Network functions are built by alternating matrix-vector multiplications with the action of the nonlinear activation function $\sigma$.

Let $\|\mathbf{W}_j\|_0$ and $|\mathbf{v}_j|_0$ be the number of nonzero entries of $\mathbf{W}_j$ and $\mathbf{v}_j$ in the $j^{\text{th}}$ hidden layer. The final form of neural network we consider in this paper is given by

$$\mathcal{F}(L, \mathbf{p}, \mathcal{N}) := \left\{ \tilde{f} \text{ of the form (1)} : \sum_{j=1}^{L} \|\mathbf{W}_j\|_0 + |\mathbf{v}_j|_0 \leq \mathcal{N} \right\}. \tag{2}$$

The main advantage of using this notation comes from its convenience for tracking the construction process of network $\tilde{f}$ for approximating $f \in W_\infty^r(\mathcal{S}^{d-1})$. See Section D.2 in the Appendix. Now, we define the Sobolev spaces over the sphere in the next Subsection.

### 2.2 DEFINITION OF SOBOLEV SPACES OVER SPHERE

For $1 \leq p \leq \infty$, we denote $\mathcal{L}_p(\mathcal{S}^{d-1}) = \mathcal{L}_p(\mathcal{S}^{d-1}, \rho_{\mathcal{X}})$ as the $\mathcal{L}_p$-function space defined with respect to the normalized Lebesgue measure $\rho_{\mathcal{X}}$ on $\mathcal{S}^{d-1}$, with norm $\|g\|_p := \left( \int_{\mathcal{S}^{d-1}} |g(\mathbf{x})|^p \rho_{\mathcal{X}}(d\mathbf{x}) \right)^{1/p}$.

Let $\mathcal{H}_k^d$ be the space of homogeneous harmonic polynomials of total degree $k \in \mathbb{Z}_+$ restricted on $\mathcal{S}^{d-1} \subset \mathbb{R}^d$. In Dai & Xu (2013); Efthimiou & Frye (2014), its dimension for $k \in \mathbb{N}$ is found to be

$$\mathcal{N}(k, d) = \frac{2k + d - 2}{k} \binom{k + d - 3}{k - 1}. \tag{3}$$

Note that $\mathcal{L}_2(\mathcal{S}^{d-1})$ is a Hilbert space with inner product $\langle f, g \rangle_{\mathcal{L}_2(\mathcal{S}^{d-1})} := \int_{\mathcal{S}^{d-1}} f(\mathbf{x})g(\mathbf{x})\rho_{\mathcal{X}}(\mathbf{x})$ for $f, g \in \mathcal{L}_2(\mathcal{S}^{d-1})$. The spaces $\mathcal{H}_k^d$, for $k \in \mathbb{Z}_+$, of spherical harmonics are mutually orthogonal with respect to the inner product of $\mathcal{L}_2(\mathcal{S}^{d-1})$.

Since the space of spherical polynomials is dense in $\mathcal{L}_2(\mathcal{S}^{d-1})$, every $f \in \mathcal{L}_2(\mathcal{S}^{d-1})$ has a spherical harmonic expansion $f = \sum_{k=0}^{\infty} \mathbf{Proj}_k(f) = \sum_{k=0}^{\infty} \sum_{\ell=1}^{\mathcal{N}(k,d)} \widehat{f}_{k,\ell} \mathbf{Y}_{k,\ell}$ converging in the $\mathcal{L}_2(\mathcal{S}^{d-1})$ norm. Hereafter, $\{\mathbf{Y}_{k,\ell}\}_{\ell=1}^{\mathcal{N}(k,d)}$ denotes an orthonormal basis of $\mathcal{H}_k^d$, $\widehat{f}_{k,\ell}$ is the Fourier coefficients of $f$ given by $\widehat{f}_{k,\ell} := \langle f, \mathbf{Y}_{k,\ell} \rangle_{\mathcal{L}_2(\mathcal{S}^{d-1})} := \int_{\mathcal{S}^{d-1}} f(\mathbf{x})\mathbf{Y}_{k,\ell}(\mathbf{x})\rho_{\mathcal{X}}(d\mathbf{x})$, and $\mathbf{Proj}_k(f)$ denotes the orthogonal projection of $\mathcal{L}_2(\mathcal{S}^{d-1})$ onto $\mathcal{H}_k^d$, which has an integral representation $\mathbf{Proj}_k(f)(\mathbf{x}) := \int_{\mathcal{S}^{d-1}} f(\mathbf{y})\mathcal{Z}_k(\mathbf{x}, \mathbf{y})\rho_{\mathcal{X}}(d\mathbf{y})$, $\forall \mathbf{x} \in \mathcal{S}^{d-1}$, where $\mathcal{Z}_k(\mathbf{x}, \mathbf{y}) := \sum_{\ell=1}^{\mathcal{N}(k,d)} \mathbf{Y}_{k,\ell}(\mathbf{x})\mathbf{Y}_{k,\ell}(\mathbf{y})$, $\forall \mathbf{x}, \mathbf{y} \in \mathcal{S}^{d-1}$.

We know that $\mathcal{Z}_k(\mathbf{x}, \mathbf{y})$ is a reproducing kernel of $\mathcal{H}_k^d$, independent of the choice of $\{\mathbf{Y}_{k,\ell}\}_{\ell=1}^{\mathcal{N}(k,d)}$, and with $\lambda_G = \frac{d-2}{2}$, $\mathcal{Z}_k(\mathbf{x}, \mathbf{y}) := \frac{N+\lambda_G}{\lambda_G} \mathcal{G}_k^{\lambda_G}(\langle \mathbf{x}, \mathbf{y}\rangle)$, $\quad \forall \mathbf{x}, \mathbf{y} \in \mathcal{S}^{d-1}$ where $\mathcal{G}_k^{\lambda_G}$ is the Gegenbauer polynomial of degree $k$ with parameter $\lambda_G > -\frac{1}{2}$, see for instance Dai & Xu (2013). Denote $u := \langle \mathbf{x}, \mathbf{y}\rangle$, the exact expression of $\mathcal{G}_k^{\lambda_G}(u)$ is given in terms of the Gamma function by

$$\mathcal{G}_k^{\lambda_G}(u) := \sum_{\ell=0}^{\lfloor \frac{k}{2}\rfloor} (-1)^\ell \frac{\Gamma(k-\ell+\lambda_G)}{\Gamma(\lambda_G)\ell!(k-2\ell)!}(2u)^{k-2\ell}. \tag{4}$$

The space of $\mathcal{H}_k^d$ of spherical harmonics can also be characterized as eigenfunction spaces of the Laplace-Beltrami operator $\Delta_{\mathcal{S}^{d-1}}$ on $\mathcal{S}^{d-1}$. Indeed, $\mathcal{H}_k^d = \left\{ f \in \mathcal{C}^2(\mathcal{S}^{d-1}) : \Delta_{\mathcal{S}^{d-1}}f = -\lambda_k f \right\}$, where $\lambda_k = k(k+d-2)$ and $\mathcal{C}^2(\mathcal{S}^{d-1})$ denotes the space of all twice continuously differentiable functions on $\mathcal{S}^{d-1}$. In fact, with the identity operator $\mathcal{I}$, we may define the fractional power of $(-\Delta_{\mathcal{S}^{d-1}} + \mathcal{I})^\alpha$ of the operator $(-\Delta_{\mathcal{S}^{d-1}} + \mathcal{I})$ in a distributional sense for $\alpha \in \mathbb{R}$: $\mathbf{Proj}_k((-\Delta_{\mathcal{S}^{d-1}} + \mathcal{I})^\alpha f) = (1+\lambda_k)^\alpha \mathbf{Proj}_k(f)$. Now, we define the Sobolev space $W_p^r(\mathcal{S}^{d-1})$ to be the subspace of $\mathcal{L}_p(\mathcal{S}^{d-1})$ for $1 \leq p \leq \infty, r > 0$, with the finite norm

$$\|f\|_{W_p^r(\mathcal{S}^{d-1})} = \left\| (-\Delta_{\mathcal{S}^{d-1}} + \mathcal{I})^{r/2} f \right\|_p < \infty. \tag{5}$$

In this paper, we consider the case $p = \infty$ (i.e., $f \in W_\infty^r(\mathcal{S}^{d-1})$), which is essentially the Hölder space. The sphere $\mathbb{S}^{d-1}$ is a smooth Riemannian manifold without boundary. Its nice Laplace-Beltrami operator (i.e., $\Delta_{\mathcal{S}^{d-1}}$) acting as a Hessian operator of functions on the sphere gives the natural definition of Sobolev spaces $W_\infty^r(\mathbb{S}^{d-1})$ in (5); that is, the Sobolev space is a collection of continuous functions defined on sphere $\mathbb{S}^{d-1}$ whose generalized (distributional) derivatives up to order $r$ are essentially bounded. See Equations (16) in Hesse (2006), (3.4) in Fang et al. (2020), (16) in Feng et al. (2021), (5.1.9) in Freeden et al. (1998) for more detailed treatments on $W_\infty^r(\mathbb{S}^{d-1})$. Readers can also refer the definition of $W_\infty^r(\mathcal{C}^d)$ in the Appendix A, when $\mathcal{C}^d = [0,1]^d$, for comparison with $W_\infty^r(\mathcal{S}^{d-1})$ and later use in Subsection 3.1.

## 3 APPROXIMATION ERROR

Now, we present our Theorem on approximating functions $f \in W_\infty^r(\mathcal{S}^{d-1})$ via $\mathcal{F}(L, \mathbf{p}, \mathcal{N})$ in (2).

**Theorem 3.1** Let $0 < \alpha < 1, m, N, M \in \mathbb{N}$ with $1 \leq N \leq d^\alpha + 1$. For any function $f \in W_\infty^r(\mathcal{S}^{d-1})$ with $r > 0$, there exists a network

$$\tilde{f} \in \mathcal{F}(L, (d, 22NM, \ldots, 22NM, 1), \mathcal{N}) \tag{6}$$

with depth $L = (m+4)\lceil \log_2(2N)\rceil$ and number of parameters $\mathcal{N} \leq M(2d + 404N \cdot (m+3) + 2N + 4) + 1$ such that

$$\left\| f - \tilde{f} \right\|_\infty \leq C_\eta'' \|f\|_{W_\infty^r(\mathcal{S}^{d-1})} \times$$

$$\max\left\{ N^{-r}, \frac{\left(\frac{6}{\pi e}\right)^{\frac{d}{4}} d^{N+\frac{3d-4r-2}{8}}(2N+1)^{\frac{3d-4r}{4}}}{\sqrt{M}}, d^{2N}\left(\log_2(2N)\right)^2 2^{-2m} \right\}, \tag{7}$$

where $C_\eta''$ is a constant dependent on $\eta$, and independent on $d, r, N, M$ or $f$.

The proof of Theorem 3.1 is lengthy and technical. We provide detailed proof ideas with technical remarks for the Lemmas and Proposition used for the proof of Theorem 3.1 in the Appendix D. The detailed technical proofs of those Lemmas and Proposition are provided in the Appendix E. Here, for conciseness, we provide some important remarks on the Theorem and a simple proof sketch, which starts with a simple triangle inequality:

$$\left\| f - \tilde{f} \right\|_\infty \leq \|f - L_N(f)\|_\infty + \left\| L_N(f) - \widehat{L}_{N,M}^{\boldsymbol{y}}(f) \right\|_\infty + \left\| \widehat{L}_{N,M}^{\boldsymbol{y}}(f) - \tilde{f} \right\|_\infty. \tag{8}$$

Three error terms in (7) correspond to the bounds on three terms of the right-hand side in the inequality (8). We want to emphasize that the constant $C_\eta'' > 0$ in (7) is independent of $d$. Furthermore, we track how the bound is explicitly dependent on $d$ allowing it to tend to infinity.

For first term, note that any $f \in W_\infty^r(\mathcal{S}^{d-1})$ is approximated by a weighted sum of $\mathbf{Proj}_k(f)$ for $0 \le k \le 2N$, denoted as $L_N(f)$. The corresponding approximation error is small for large enough $N$ and $r$. Here, importantly, we set the $N = \lceil d^\alpha \rceil$ for $0 < \alpha < 1$, so that the input dimension $d$ grows faster than $N$.

For the second term, notice that the definition of $L_N(f)$ is involved with the integral over the sphere, and the key for approximating the function is to discretize this integral by $M$ random samples $\mathbf{y} = \{\mathbf{y_1}, \ldots, \mathbf{y_M}\}$ independently drawn from $\rho_\mathcal{X}$. The discretized version of $L_N(f)$ is denoted as $\widehat{L}_{N,M}^{\mathbf{y}}(f)$. As observed in the error bound, the higher degree $N$ the $L_N(f)$ has, the more sampled points $M$ the approximation requires. However, the requirement is ameliorated as $r$ increases. A similar effect can be observed in the constant factor in $d$. For the fixed smoothness index $r$, the higher the data dimension $d$ is, the more the sampled point $M$ is required for good approximation, but the requirement is alleviated as the smoothness index $r$ increases. If $r$ increases up to order $d$, the factor [2] decays exponentially fast as $d \to \infty$, eventually letting $M \ge 1$ to be any integer. This phenomenon is further investigated in the Corollary 3.3.

The last term corresponds to the error of the neural network $\tilde{f}$ approximating $\widehat{L}_{N,M}^{\mathbf{y}}(f)$. For any point $\mathbf{x} \in \mathcal{S}^{d-1}$, the evaluated function value $\widehat{L}_{N,M}^{\mathbf{y}}(f)(\mathbf{x})$ is simply a weighted average of $\xi_{N,r}(\langle \mathbf{x}, \mathbf{y}_i \rangle)$, for the sampled $\mathbf{y} = \{\mathbf{y_1}, \ldots, \mathbf{y_M}\}$. Here, $\xi_{N,r}(\langle \mathbf{x}, \mathbf{y}_i \rangle)$ is a linear combination of $\mathcal{G}_k^{\lambda_G}(\langle \mathbf{x}, \mathbf{y}_i \rangle)$ in (4) for $0 \le k \le 2N$. Thus, it is the sum of univariate polynomials of degree up to $2N$. We construct sub-networks approximating $\xi_{N,r}(\langle \mathbf{x}, \mathbf{y}_i \rangle)$ for each $i \in [M]$. This explains the width of $\tilde{f}$ is proportional to $NM$. The corresponding error bound is dependent on $d^{2N}$, where it comes from the applications of Stirling's formula on the coefficient factors in $\mathcal{G}_k^{\lambda_G}(\langle \mathbf{x}, \mathbf{y}_i \rangle)$. The error, $\left(\log_2(2N)\right)^2 2^{-2m}$, comes from approximating $\langle \mathbf{x}, \mathbf{y}_i \rangle^k$ for $0 \le k \le 2N$ via neural networks. The larger the $m$ is, the deeper the network becomes as $L = \mathcal{O}(m)$, and the error gets smaller.

## 3.1 COMPARISON WITH SCHMIDT-HIEBER (2020)

In this Subsection, we compare the result from Theorem 3.1 with the result from Schmidt-Hieber (2020), where they consider the approximation of $f \in W_\infty^r([0,1]^d)$ via deep ReLU FNN. The Theorem is stated as follows:

**Theorem 3.2** *[Theorem 5 of Schmidt-Hieber (2020)] For any function $f \in W_\infty^r([0,1]^d)$ and let $K > 0$ be the radius of Hölder ball. Then, for any integers $m \ge 1$ and $N^H \ge (r+1)^d \vee (K+1)e^d$, there exists a network*

$$\tilde{f}^H \in \mathcal{F}^H\left(L, (d, 6(d+\lceil r \rceil)N^H, \ldots, 6(d+\lceil r \rceil)N^H, 1), \mathcal{N}^H\right) \tag{9}$$

*with depth $L = 8 + (m+5)\left(1 + \lceil \log_2(d \vee r) \rceil\right)$ and the number of parameters $\mathcal{N}^H \le 141(1 + d + r)^{3+d}N^H(m+6)$, such that*

$$\left\|f - \tilde{f}^H\right\|_\infty \le (2K+1)(1 + d^2 + r^2)6^d\left(N^H\right)2^{-m} + K3^r\left(N^H\right)^{-\frac{r}{d}}. \tag{10}$$

To avoid the confusion with the notations used in Theorem 3.1, we put the superscript $H$ to a parameter that determines width of the network (i.e., $N^H$), to the total number of parameters in the network (i.e., $\mathcal{N}^H$), and to the network class (i.e., $\mathcal{F}^H$). It is interesting to note that the exponential growth of the network size in $d$ is observed in the construction of $\mathcal{F}^H$, whereas there exists a flexibility in $\mathcal{F}$, dependent on the choice of $M$. Specifically, the width of the network in $\mathcal{F}^H$ is exponentially dependent on $d$ as $N^H = \Omega(r^d \vee e^d)$, whereas the width of the network in $\mathcal{F}$ is dependent on two parameters $N = o(d)$ and any integers $M \ge 1$. For the total number of network parameters, we have $\mathcal{N}^H = \mathcal{O}((d+r)^d)$, whereas $\mathcal{N} = \mathcal{O}(Md + Nmd)$.

---

[2]If $r = \mathcal{O}(d)$, the factor becomes $\left(\frac{6}{\pi e}\right)^{\frac{d}{4}} d^{\lceil d^\alpha \rceil}$ for $0 < \alpha < 1$. Here, the exponential decay term $\left(\frac{6}{\pi e}\right)^{\frac{d}{4}}$ is derived from Sobolev embedding Lemma. See Proposition D.1.3 in Appendix D.

Analogously, the bound on the approximation error of $\tilde{f}^H$ in (10) is dependent on $d$ exponentially, but this exponential dependence in $d$ can be avoided in the error bound of $\tilde{f}$ in (7) under two scenarios: **(I)** $r = \mathcal{O}(d)$ **and any integer** $M \geq 1$ or **(II)** $r = \mathcal{O}(1)$ **and** $M = \mathcal{O}(d^d)$. In the Corollary presented in the next Subsection, we further specify the two scenarios, and describe how the approximation error bound in each scenario converges to $0$ in terms of $d$.

### 3.2 FAST APPROXIMATION ERROR IN TERMS OF $d$

**Corollary 3.3** *Let $0 < \alpha, \beta, \gamma < 1$ with $\gamma > \max\{\alpha, \beta\}$ and $N \in \mathbb{N}$ with $1 \leq N \leq d^\alpha + 1$. For any $f \in W_\infty^r(\mathcal{S}^{d-1})$ with $r > 0$, we have:*

(I) *For $\frac{3d-2}{4} - C_1 \leq r \leq \frac{3d-2}{4}$ with some constant $C_1 \geq 0$ independent of $d$, there exists a network*

$$\tilde{f}^{(I)} \in \mathcal{F}\left(L, (d, 66N, 66N, \ldots, 66N, 1), \mathcal{N}\right)$$

*with depth $L = \mathcal{O}\left(d^\gamma \log_2 d\right)$ and the number of active parameters $\mathcal{N} = \mathcal{O}\left(d^{\max\{\alpha+\gamma, 1\}}\right)$, such that $\left\| f - \tilde{f}^{(I)} \right\|_\infty \leq C'_{\eta, \alpha, \beta, \gamma} \|f\|_{W_\infty^r(\mathcal{S}^{d-1})} d^{-d^\beta}$, where $C'_{\eta, \alpha, \beta, \gamma}$ is a constant depending only on $C_1, \eta, \alpha, \beta$ and $\gamma$.*

(II) *For $r = \mathcal{O}(1)$ and $M = \mathcal{O}\left(9^d d^{\frac{9}{4}d}\right)$, there exists a network*

$$\tilde{f}^{(II)} \in \mathcal{F}\left(L, \left(d, 22NM, \ldots, 22NM, 1\right), \mathcal{N}\right)$$

*with depth $L = \mathcal{O}\left(d^\gamma \log_2 d\right)$ and the number of active parameters $\mathcal{N} = \mathcal{O}\left(9^d d^{\frac{13}{4}d}\right)$ such that $\left\| f - \tilde{f}^{(II)} \right\|_\infty \leq C'_{\eta, \alpha, \beta, \gamma} \|f\|_{W_\infty^r(\mathcal{S}^{d-1})} d^{-\alpha r}$, where $C'_{\eta, \alpha, \beta, \gamma}$ is a constant depending only on $\eta, \alpha, \beta$ and $\gamma$.*

The detailed proof on Corollary 3.3 is deferred in the Appendix E.6. The approximation error in scenario (I) decays at a rate $d^{-d^\beta}$ for $0 < \beta < 1$, while the required number of active parameters $\mathcal{N}$ is at most $\mathcal{O}(d^2)$. Here, the construction of network $\tilde{f}^{(I)}$ is independent with the choice of $M$, and we simply choose $M = 3$. In scenario (II), since $r = \mathcal{O}(1)$ and $0 < \alpha < 1$, the approximation error decays to $0$ at $d^{-\mathcal{O}(1)}$ rate, which can be slower than $d^{-d^\beta}$ for $\beta$ close to 1. The width of $\tilde{f}^{(II)}$ grows exponentially in $d$ requiring $M = \mathcal{O}(d^d)$. Interestingly, in both scenarios, the depth $L$ has the same order in $d$ as $\mathcal{O}\left(d^\gamma \log_2 d\right)$ for $0 < \gamma < 1$.

**Remark 3.4** *As suggested by one of the reviewers, we further compare our results in Corollary 3.3 (I) with the CNN architecture with downsampling operation suggested in Fang et al. (2020) for approximating $f \in W_\infty^r(\mathcal{S}^{d-1})$, and (II) with Lu et al. (2021); Jiao et al. (2021) where they consider the problem of approximating $f \in W_\infty^r(\mathcal{C}^d)$ via deep ReLU FNN. Due to the limited space, we defer the detailed remarks on the comparisons in the Appendix C.*

## 4 STATISTICAL RISK BOUND

Let $\mathcal{X} := \mathcal{S}^{d-1}$ and $\mathcal{Y} \subset \mathbb{R}$ be the measureable feature space and output space. We denote $\rho$ as a joint probability measure on the product space $\mathcal{Z} := \mathcal{X} \times \mathcal{Y}$, and let $\rho_{\mathcal{X}}$ be the marginal distribution of the feature space $\mathcal{X}$. We assume that the noisy data set $\mathcal{D} := \{(\mathbf{x}_i, \mathbf{y}_i)\}_{i=1}^n$ are generated from the non-parametric regression model

$$\mathbf{y}_i = f_\rho(\mathbf{x}_i) + \varepsilon_i, \quad i = 1, 2, \ldots, n, \tag{11}$$

where the noise $\varepsilon_i$ is assumed to be centered sub-gaussian random variable and $\mathbb{E}(\varepsilon_i|\mathbf{x}_i) = 0$. Our goal is to estimate the regression function $f_\rho(\mathbf{x})$ with the given noisy data set $\mathcal{D}$. Specifically, it is assumed that the regression function belongs to Sobolev space on $d$-dimensional sphere; that is $f_\rho \in W_\infty^r(\mathcal{S}^{d-1})$. It is easy to see regression function $f_\rho := \mathbb{E}(\mathbf{y}|\mathbf{x})$ is a minimizer of the following population risk $\mathcal{E}(f)$ defined as:

$$\mathcal{E}(f) = \mathbb{E}_{(\mathbf{x}, \mathbf{y}) \sim \rho}\left[\left(\mathbf{y} - f(\mathbf{x})\right)^2\right].$$

However, since the joint distribution $\rho$ is unknown, we cannot find $f_\rho$ directly. Instead, we solve the following empirical risk minimization problem induced from the dataset $\mathcal{D}$:

$$\widehat{f}_n = \underset{f \in \mathcal{F}(L, \mathbf{p}, \mathcal{N})}{\arg\min} \ \mathcal{E}_D(f) := \underset{f \in \mathcal{F}(L, \mathbf{p}, \mathcal{N})}{\arg\min} \left\{ \frac{1}{n} \sum_{i=1}^{n} \left( \mathbf{y}_i - f(\mathbf{x}_i) \right)^2 \right\}. \tag{12}$$

Note that the function estimator is taken from the feedforward neural network hypothesis space $\mathcal{F}(L, \mathbf{p}, \mathcal{N})^3$ defined in (2), and we denote the empirical minimizer of (12) as $\widehat{f}_n$. It is assumed that $|\mathbf{y}| \leq B$ almost everywhere and we have $|f_\rho(\mathbf{x})| \leq B$. We project the output function $f : \mathcal{S}^{d-1} \rightarrow \mathbb{R}$ onto the interval $[-B, B]$ by a projection operator

$$\pi_B f(\mathbf{x}) = \begin{cases} f(\mathbf{x}), & \text{if } -B \leq f(\mathbf{x}) \leq B, \\ B, & \text{if } f(\mathbf{x}) > B, \\ -B, & \text{if } f(\mathbf{x}) < -B. \end{cases} \tag{13}$$

We consider the clipped estimator $\pi_B \widehat{f}_n$ for recovering the regression function $f_\rho$. Note that the clipped estimator has been widely used in statistical learning papers Suzuki (2018); Fang & Cheng (2023); Oono & Suzuki (2019). The quality of $\pi_B \widehat{f}_n$ is measured through the difference between two expected risks (i.e., excess risk) defined as $\mathcal{E}\big(\pi_B \widehat{f}_n\big) - \mathcal{E}\big(f_\rho\big)$.

## 4.1 Upper-bound on excess risk

In this Subsection, we provide the upper-bound on the excess risk of the clipped estimator $\pi_B(\widehat{f}_n)$ with respect to the pseudo-dimension (i.e., $\text{Pdim}(\mathcal{F})$) and the approximation error (i.e., $\|f - f_\rho\|_\infty$). Before presenting the bound, the definition of $\text{Pdim}(\mathcal{F})$ is presented.

**Definition 4.1** *Denote by $\text{Pdim}(\mathcal{F})$, the pseudo-dimension of $\mathcal{F}$, which is the largest integer $\ell$, for which there exists $(\xi_1, \ldots, \xi_\ell, \eta_1, \ldots, \eta_\ell) \in \mathcal{X}^\ell \times \mathbb{R}^\ell$ such that for any $(a_1, \ldots, a_\ell) \in \{0,1\}^\ell$, there is some $f \in \mathcal{F}$ satisfying*

$$\forall i : f(\xi_i) > \eta_i \iff a_i = 1.$$

For more comprehensive exploration on $\text{Pdim}(\mathcal{F})$ can be found in references Anthony & Bartlett (1999); Bartlett et al. (2019). We provide the first theorem on the excess risk.

**Proposition 4.2** *Set $\delta \in (0, 1)$. Then, with probability at least $1 - \delta$, we have*

$$\mathcal{E}\big(\pi_B \widehat{f}_n\big) - \mathcal{E}\big(f_\rho\big) \leq C_{B, \delta, f} \cdot \left( \frac{\text{Pdim}(\mathcal{F}) \cdot \log(n)}{n} + \frac{\|f - f_\rho\|_\infty}{\sqrt{n}} + \|f - f_\rho\|_\infty^2 \right), \tag{14}$$

*where $C_{B, \delta, f}$ is an absolute constant dependent on $B, \delta, f$ independent on $n, r, d$.*

A detailed proof of Proposition 4.2 is deferred in the Appendix. The excess risk $\mathcal{E}\big(\pi_B \widehat{f}_n\big) - \mathcal{E}\big(f_\rho\big)$ is a random quantity over the estimator $\widehat{f}_n$ and the statement in the Theorem holds with probability at least $1 - \delta$. The failure probability $\delta \in (0, 1)$ is hidden in the constant $C_{B, \delta, f}$ logarithmically, i.e., $\log(\frac{1}{\delta})$. In the bound, it should be noted that there is a trade-off between the ***"approximation error"*** (i.e., $\|f - f_\rho\|_\infty$) term and the combinatorial ***"complexity measure"*** term of a neural network class $\mathcal{F}$ (i.e., $\text{Pdim}(\mathcal{F}) \cdot \log(n)/n$); that is, the richer the network hypothesis space $\mathcal{F}$ becomes, the finer the approximation result we get. Nonetheless, the arbitrary increase in the hypothesis space $\mathcal{F}$ eventually leads to the increase of the bound in excess risk. In the following Subsection, we will show how the specifications (i.e., the choices of $(L, \mathbf{p}, \mathcal{N})$) of the network architecture affect the tension between these two terms.

## 4.2 Convergence Rate of Excess Risk

Now we are ready to formally state bounds on the excess risks of $\pi_M \widehat{f}_n$ when $f_\rho \in W_\infty^r(\mathcal{S}^{d-1})$ (i.e., Theorem 4.3) and $f_\rho \in W_\infty^r([0, 1]^d)$ (i.e., Theorem 4.4), respectively.

---

[3]Henceforth, we will use a shorthand notation of $\mathcal{F}(L, \mathbf{p}, \mathcal{N})$ as $\mathcal{F}$. Dependence on $(L, \mathbf{p}, \mathcal{N})$ should be implicitly understood.

**Theorem 4.3** *Suppose $f_\rho \in W_\infty^r(\mathcal{S}^{d-1})$ with $r > 0$. A network $\widehat{f}_n$ from (6) with choices $N = \lceil n^{\frac{2}{3d+4r}} \rceil$, $M = \lceil n^{\frac{3d}{3d+4r}} \rceil$, and $m = \lceil \frac{r}{3d+4r} \log_2(n) \rceil$ yield the bound on the excess risk with probability at least $1 - \delta$ as follows:*

$$\mathcal{E}(\pi_M \widehat{f}_n) - \mathcal{E}(f_\rho)$$

$$\leq \mathcal{C}_{B,\eta,\delta,f} \cdot \max\left\{1, \frac{6rd}{(3d+4r)^2}(\log_2(n))^4, \left(\frac{6}{\pi e}\right)^{\frac{d}{2}} d^{2N+\frac{3d-4r-2}{4}}, d^{4N}\right\} \cdot n^{-\frac{2r}{2r+1.5d}}, \qquad (15)$$

*where $\mathcal{C}_{B,\eta,\delta,f}$ depends on $B$, $\eta$, $\delta$, $f$ and independent on $d, r$ and $n$.*

**Theorem 4.4** *Suppose $f_\rho \in W_\infty^r([0,1]^d)$ with $r > 0$. A network $\widehat{f}_n$ from (9) with choices $N^H = \lceil n^{\frac{d}{2d+r}} \rceil$, and $m^H = \lceil \frac{d+r}{d+2r} \log_2(n) \rceil$ yield the bound on the excess risk with probability at least $1 - \delta$ as follows:*

$$\mathcal{E}(\pi_M \widehat{f}_n) - \mathcal{E}(f_\rho) \qquad\qquad\qquad\qquad\qquad\qquad\qquad\qquad\qquad (16)$$

$$\leq \mathcal{C}_{B,\eta,\delta,K} \cdot \max\left\{\lceil(\log_2((d+\lceil r \rceil)n^2)\rceil^2(d+r)^d \cdot (\log_2(n))^3, \left(1+r^2+d^2\right)^2 6^{2d} + 3^{2r}\right\} \cdot n^{-\frac{2r}{2r+d}},$$

*where $\mathcal{C}_{B,\eta,\delta,K}$ depends on $B$, $\eta$, $\delta$, $K$ and independent on $d, r$ and $n$.*

Detailed proofs on Theorems 4.3 and 4.4 are deferred in the Appendix F.2 and F.3. Both proofs are simple applications of Proposition 4.2 with results from Theorem 3.1 and 3.2. For both cases, Pdim($\mathcal{F}$) can be easily computed from Lemma H.1 in the Appendix. The parameters that determine the network architectures, $N, M, m$ and $N^H, m^H$ in two Theorems are chosen in a way that the bound in (14) is tight in terms of sample size $n$. Constant factors $\mathcal{C}_{B,\eta,\delta,f}$ and $\mathcal{C}_{B,\eta,\delta,K}$ are dependent on $\delta \in (0,1)$ as $\log(\frac{1}{\delta})$. The bound in Theorem 4.3, $\mathcal{O}_d(n^{-\frac{2r}{2r+1.5d}})$, is sub-optimal[4] in a minimax sense for estimating functions $f_\rho \in W_\infty^r(\mathcal{S}^{d-1})$, where $\mathcal{O}_d$ hides the constant factor in $d$. The extra $0.5d$ factor in the denominator of exponent comes from the Sobolev embedding Lemma (Lemma D.1.3) and discretization Lemma (Lemma D.1.4). For the constant factor in $d$, when $r = \mathcal{O}(1)$, the exponential dependence on $d$ can be observed. However, when $r = \mathcal{O}(d)$, the excess bound in (15) reduces to $\mathcal{E}(\pi_M \widehat{f}_n) - \mathcal{E}(f_\rho) \leq \mathcal{C}_{B,\eta,\delta,f} \cdot \max\left\{(\log_2(n))^4, d^{4N}\right\} \cdot n^{-\frac{2r}{2r+1.5d}}$. With a choice of $N = \lceil n^{\frac{2}{3d+4r}} \rceil$, as $d, r \to \infty$, the constant $d^{4N}$ becomes $d^{\mathcal{O}(1)}$. In contrast, in (16) for estimating functions $f_\rho \in W_\infty^r([0,1]^d)$, the rate $n^{-\frac{2r}{2r+d}}$ is minimax optimal, but we cannot observe the interactions between $r$ and $d$ as we observe in (15).

**Remark 4.5** *From the technical point of view, the result in Theorem 4.3 should be compared with the results in the existing literature, i.e., Schmidt-Hieber (2020); Chen et al. (2022); Suzuki (2018), in a sense that our result doesn't require the boundedness of the weight parameters in the network construction. The detailed readings of their proofs reveal that they require the bound on the uniform covering number of $\mathcal{F}$ and it can be bounded by the Lipschitzness of the network output with respect to the weight parameters. Naturally, for the discretizations of the parameter space, the boundedness assumption is required. In contrast, in our result, due from the Bartlett et al. (2019) (See Lemma H.1), bounding the complexity measure Pdim($\mathcal{F}$) doesn't require the parameter boundedness assumption.*

## 5 An Open Question

In this paper, we prove when $r = \mathcal{O}(d)$, deep ReLU FNNs only require at most $\mathcal{N} = \mathcal{O}(d^2)$ parameters to get a sharp approximation rate. However, this condition seems restrictive, and needs further investigation whether it is a necessary and sufficient condition to avoid the curse of dimensionality for approximating $f \in W_\infty^r(\mathcal{S}^{d-1})$. To answer this question, it is essential to study the lower bound of $\mathcal{N}$ with a similar approximation error as stated in Theorem 3.1, and see if it has the matching order with the upper-bound we get in $d$. We conjecture obtaining this result is possible by combining the ideas of using VC-dimension of deep ReLU FNNs (Bartlett et al., 2019; Yarotsky, 2017) and of constructing the packing set on the sphere through the spherical cap (Hesse, 2006), while tracking the $d$-dependency in the constant factor carefully. We leave this for future research.

---

[4]Since $\mathcal{S}^{d-1} \subset \mathcal{C}^d$, it seems obvious we should achieve minimax optimal rate. In this regard, we add further detailed technical remarks on the sub-optimality of excess risk in the Appendix B.

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

# A $d^d$-DEPENDENT CONSTANT IN $\mathcal{N}$ FOR APPROXIMATING $f \in W_\infty^r([0,1]^d)$

First, we define the function space $W_\infty^r([0,1]^d)$ on the $d$-dimensional unit cube. For $r = n + \sigma$ where $n \in \mathbb{N}_0$ and $\sigma \in (0,1]$, a function has Hölder smoothness index $r$ if all partial derivatives up to order $n$ exist and are bounded and the partial derivatives of order $n$ are $\sigma$ Hölder. Formally, the ball of $r$-Hölder functions with radius $\mathcal{Q}$ is then defined as

$$W_\infty^r([0,1]^d) = \left\{ f : [0,1]^d \to \mathbb{R} : \sum_{\boldsymbol{\alpha}:|\boldsymbol{\alpha}|\leq n} \|\partial^{\boldsymbol{\alpha}} f\|_\infty + \sum_{\boldsymbol{\alpha}:|\boldsymbol{\alpha}|=n} \sup_{\substack{\boldsymbol{x},\boldsymbol{y}\in[0,1]^d \\ \boldsymbol{x}\neq\boldsymbol{y}}} \frac{|\partial^{\boldsymbol{\alpha}} f(\boldsymbol{x}) - \partial^{\boldsymbol{\alpha}} f(\boldsymbol{y})|}{|\boldsymbol{x}-\boldsymbol{y}|_\infty^\sigma} \leq \mathcal{Q} \right\}.$$

where $\partial^{\boldsymbol{\alpha}} f := \frac{\partial^{|\boldsymbol{\alpha}|}}{\partial^{\alpha_1}\ldots\partial^{\alpha_d}} f$ for the multi-index notation, $\boldsymbol{\alpha} := (\alpha_1,\ldots,\alpha_d)$.

The fundamental ideas for approximating functions $f \in W_\infty^r([0,1]^d)$ in the existing literature rely on a local Taylor approximation technique. The technique discretizes $d$-dimensional input cube into a sub-cube set whose size is $(K+1)^d$ where $(K+1)$ is the grid size of each coordinate. For any $\mathbf{x}$ in the input cube, the function $f$ is approximated by using the closest $2^d$ grid points to $\mathbf{x}$ via Taylor expansion of $f$ up to the degree $\lfloor r \rfloor$, where we denote the largest integer less than or equal to $u > 0$ as $\lfloor u \rfloor$. Therefore, the total number of active parameters for the net is at least more than the total number of coefficients of partial derivatives $\partial^\alpha f := \frac{\partial^{|\alpha|}}{\partial^{\alpha_1}\ldots\partial^{\alpha_d}} f$ for $|\alpha| = |\alpha_1| + \cdots + |\alpha_d| \leq \lfloor r \rfloor$. This yields the lower bound on the active parameters for the network via local Taylor approximation as $(K+1)^d \cdot \sum_{i=0}^{\lfloor r \rfloor} \binom{d+i-1}{d}$.

# B SUBOPTIMAL CONVERGENCE RATE OF EXCESS RISK FOR $f \in W_\infty^r(\mathcal{S}^{d-1})$

Since $\mathcal{S}^{d-1} \subset \mathcal{C}^d$, in light of Whitney's extension theorem, it is obvious that the convergence rate of excess risk for $f \in W_\infty^r(\mathcal{S}^{d-1})$ should achieve the minimax optimal rate $n^{-\frac{2r}{2r+d}}$ same as for estimating $f \in W_\infty^r(\mathcal{C}^d)$. However, from this perspective, it is not possible to track the explicit dependence on $d$ in prefactor of the rate.

From Proposition 4.2, we track this dependence with the combination of our own approximation result. First of all, in order to achieve the minimax learning rate in $n$ (e.g., sample size); that is, $n^{-\frac{2r}{2r+d}}$, we need to achieve the optimal approximation rate, known as $\mathcal{O}(\mathcal{N}^{-\frac{r}{d}})$, where $\mathcal{N}$ denotes the number of active parameters in the network. This can be easily checked in Schmidt-Hieber (2020)'s result.

But this cannot be achieved in our Theorem 3.1, the result of approximation theorem. The main reason arises from the employment of Sobolev embedding theorem (Proposition D.1.3) and of concentration inequality from Smale & Zhou (2007) (Lemma E.2.1) used for bounding the term $\|L_N(f) - \widehat{L}_{N,M}^{\boldsymbol{y}}(f)\|_\infty$. Proposition D.1.3 specifies $\|L_N(f) - \widehat{L}_{N,M}^{\boldsymbol{y}}(f)\|_\infty \leq C_d\|L_N(f) - \widehat{L}_{N,M}^{\boldsymbol{y}}(f)\|_{W_2^s(\mathcal{S}^{d-1})}$ for some $s > (3d-2)/4$ and Lemma D.1.4 specifies the relations $\|L_N(f) - \widehat{L}_{N,M}^{\boldsymbol{y}}(f)\|_{W_2^s(\mathcal{S}^{d-1})} \leq C'_{d,r}\|f\|_{W_\infty^r(\mathcal{S}^{d-1})} \frac{(2N+1)^{\frac{3d-4r}{4}}}{\sqrt{M}}$. In this process, there are some extra factors multiplied leading to the rate sub-optimal, reflected in the term $(2N+1)^{\frac{3d-4r}{4}}$. It would be an interesting research direction, if we can develop a nice mathematical framework eliminating this extra factor.

# C FURTHER COMPARISONS WITH EXISTING LITERATURE

## C.1 COMPARISONS OF COROLLARY 3.3. WITH THE CNN APPROXIMATOR FOR THE APPROXIMATION OF $f \in W_\infty^r(\mathcal{S}^{d-1})$

We consider the case when the approximator is deep ReLU CNN followed by downsampling operations and very few fully-connected layers. This is exactly the same architecture considered in the paper Fang et al. (2020), and by applying our result (Lemma D.1.4), we get the following theorem.

**Theorem C.1.** *Let* $2 \leq S \leq d$, $0 < \alpha < 1$, *and* $B, N, M \in \mathbb{N}$ *with* $1 \leq N \leq d^\alpha + 1$. *Let* $J \geq \lceil \frac{Md-1}{S-1} \rceil$, $\mathcal{D}_1 = (2B+3)\lfloor(d+JS)/d\rfloor$, *and* $\mathcal{D}_2 = \lfloor(d+JS)/d\rfloor$. *Then, for any function* $f \in$

|  | Schmidt-Hieber (Thm 5) | Lu et al. (Thm 1.1.) | Jiao et al. (Corol. 3.1) |
|---|---|---|---|
| Depth | $\mathcal{O}(\lceil \log_2(r \vee d) \rceil)$ | $\mathcal{O}(r^2 + d)$ | $\mathcal{O}((\lfloor r \rfloor + 1)^2)$ |
| Width | $\mathcal{O}((d + \lceil r \rceil)(r+1)^d)$ | $\mathcal{O}(r^{d+1} d 3^d)$ | $\mathcal{O}((\lfloor r \rfloor + 1)^2 3^d d^{\lfloor r \rfloor + 1})$ |
| Approx. Rate | $\mathcal{O}(6^d(r+1)^d)$ | $\mathcal{O}(8^r(r+1)^d)$ | $\mathcal{O}((\lfloor r \rfloor + 1)^2 d^{\lfloor r \rfloor + (r \vee 1)/2})$ |

Table 2: A summary of $r$ and $d$ dependences in the prefactors of constructed deep ReLU networks' depths, widths and approximation rates for approximating the functions $f \in W_\infty^r(\mathcal{C}^d)$ in Schmidt-Hieber (2020); Lu et al. (2021); Jiao et al. (2021).

$W_\infty^r(\mathcal{S}^{d-1})$ *with* $r > 0$*, there exists a network* $\tilde{f}^{CNN} \in \mathcal{H}_{J,\mathcal{D}_1,\mathcal{D}_2,S}$ *with the number of parameters* $\mathcal{N} \leq J(3S+2) + M + 2B + 4$ *such that*

$$\left\| f - \tilde{f}^{CNN} \right\|_\infty \leq C''_{\eta,\alpha} \|f\|_{W_\infty^r(\mathcal{S}^{d-1})} \max \left\{ N^{-r}, \frac{\left(\frac{6}{\pi e}\right)^{\frac{d}{4}} d^{N + \frac{3d-4r-2}{8}} (2N+1)^{\frac{3d-4r}{4}}}{\sqrt{M}}, d^{2N} \frac{r}{r-1} \frac{N^2}{B} \right\},$$

(17)

*where* $C''_{\eta,\alpha}$ *is a constant dependent on* $\eta, \alpha$*, and independent on* $d, r, N, M$ *or* $f$*.*

The only part we need to pay attention to is bounding the term $\|\widehat{L}^{\mathbf{y}}_{N,M}(f) - \tilde{f}^{CNN}\|_\infty$, and track the explicit dependence on $d$ in the bound, and its proof is deferred in Appendix G.

We consider the case $r = \mathcal{O}(d)$ **and any integer** $M \geq 1$. In this case, our result for deep ReLU FNN shows that $\mathcal{O}(d^{-d^\beta})$ can be achieved for the approximation, with at most $\mathcal{O}(d^2)$ active parameters. In light of the result from Zhou (2020), we also should expect the same results for CNN with downsampling operation.

In order to get an approximation rate $\mathcal{O}(d^{-d^\beta})$ for some $0 < \beta < 1$, controlling the first two terms in (1.1) is the same as that of our proof in the Appendix E.6. We only need to pay attention to the last term. Since $1 \leq N \leq d^\alpha + 1$, for some $0 \leq \alpha < 1$, we have

$$\frac{r}{r-1} \cdot d^{2N} N^2 / B \leq 8 \cdot d^{2d^\alpha + 2} d^{2\alpha} / B \leq 8 \cdot d^{2d^\alpha + 4} / B \leq C \cdot d^{-d^\beta},$$

for some constant $C > 0$ independent with $d > 0$ and $r > 0$. The rate $d^{-d^\beta}$ is obtainable only when $B = \mathcal{O}(d^d)$. Then, the number of parameters $\mathcal{N} \leq J(3S+2) + M + 2B + 4$ is bounded by $\mathcal{O}(d^d)$. This is an unsatisfactory result. However, we firmly believe this result can be improved, and leave it as an open question for future work.

## C.2 COMPARISONS OF COROLLARY 3.3. WITH THE RECENT APPROXIMATION RESULTS FOR $f \in W_\infty^r(\mathcal{C}^d)$ VIA DEEP ReLU FNN

After the publication of Schmidt-Hieber (2020), a series of works further studied the approximation and estimation of $f \in W_\infty^r(\mathcal{C}^d)$ via deep ReLU FNNs. Specifically, Ghorbani et al. (2020) pointed out that the additive function studied in Schmidt-Hieber (2020) has an exponential dependence in $d$ in the prefactor of the estimation error, requiring $n \gtrsim d^d$ sample sizes for good convergence rate. Later, Shen et al. (2021) and Lu et al. (2021) tracked the explicit dependence on $d$ in the approximation error as well as in the architectural components, (i.e., width ($\mathcal{W}$) and depth ($L$)), for approximating $f \in C(\mathcal{C}^d)$ (i.e., Lipschitz continuous functions on $\mathcal{C}^d$) and $f \in W_\infty^r(\mathcal{C}^d)$, respectively. Specifically, Lu et al. (2021) improved the prefactor $\mathcal{O}(8^r(r+1)^d)$ when it is compared with $\mathcal{O}(6^d(r+1)^d)$ from Schmidt-Hieber (2020) for the approximation error. Most recently, Jiao et al. (2021) further improved the prefactor $\mathcal{O}((\lfloor r \rfloor + 1)^2 d^{\lfloor r \rfloor + (r \vee 1)/2})$ for approximating $f \in W_\infty^r(\mathcal{C}^d)$. In Table 2, comparisons of depths, widths, and approximation rates of the constructed deep ReLU networks in Schmidt-Hieber (2020); Lu et al. (2021); Jiao et al. (2021) are summarized.

Since $\mathcal{N}$ is not tracked in the works of Lu et al. (2021); Jiao et al. (2021), direct comparisons of their results with our results for the approximation of $f \in W_\infty^r(\mathcal{S}^{d-1})$ can be challenging. However, it can be roughly checked that the result in Jiao et al. (2021) doesn't escape the curse for all $r > 0$. Specifically, for $r = \mathcal{O}(1)$, the constructed network has the exponential dependence in $d$ for its width. For the case $r = \mathcal{O}(d)$, achieving the approximation rate $\mathcal{O}(d^{-d^\beta})$ requires $\mathcal{O}(d^d)$ width in the suggested network.

# D    ROADMAPS FOR PROOF OF THEOREM 3.1

In this section of the Appendix, we provide the definitions of $L_N(f)$ and $\widehat{L}_{N,M}^{\boldsymbol{y}}(f)$ along with the overall picture for the proof of Theorem 3.1. Recall we have the following decomposition:

$$\left\| f - \tilde{f} \right\|_\infty \le \underbrace{\| f - L_N(f) \|_\infty}_{:=\textbf{(I)}} + \underbrace{\left\| L_N(f) - \widehat{L}_{N,M}^{\boldsymbol{y}}(f) \right\|_\infty}_{:=\textbf{(II)}} + \underbrace{\left\| \widehat{L}_{N,M}^{\boldsymbol{y}}(f) - \tilde{f} \right\|_\infty}_{:=\textbf{(III)}}. \tag{18}$$

In Subsection D.1, we provide the idea for bounding **(I)** and **(II)**. In Subsection D.2, the construction of neural network $\tilde{f}$ for approximating $\widehat{L}_{N,M}^{\boldsymbol{y}}(f)$ is described. In this section, no proofs of Propositions and Lemmas are included, but only key ideas for the proofs and technical comparisons with other literature are provided. All the detailed proofs of technical statements in this section are deferred in the Appendix $C.1$.

## D.1    ERROR BOUNDS FOR **(I)** AND **(II)**

A function $f \in W_\infty^r(\mathcal{S}^{d-1})$ is approximated by a linear scheme $L_N$ defined as follows.

**Definition D.1.1** *Given a $C^\infty([0,\infty])$ function $\eta$ with $\eta(t) = 1$ for $0 \le t \le 1$ and $\eta(t) = 0$ for $t \ge 2$, we define a sequence of linear operator $L_N$, $N \in \mathbb{N}$, on $\mathcal{L}_p(\mathcal{S}^{d-1})$ with $1 \le p \le \infty$ by*

$$L_N(f)(\mathbf{x}) := \sum_{k=0}^{2N} \eta\left(\frac{k}{N}\right) \boldsymbol{Proj}_k(f)(\mathbf{x}) = \int_{\mathcal{S}^{d-1}} f(\mathbf{y}) \ell_{k,d}(\langle \mathbf{x}, \mathbf{y} \rangle) \rho_\mathcal{X}(d\mathbf{y}), \quad \mathbf{x} \in \mathcal{S}^{d-1}, \tag{19}$$

*where with $\lambda_G = \frac{d-2}{2}$, $\ell_{N,d}$ is a kernel given by*

$$\ell_{N,d}(t) := \sum_{k=0}^{2N} \eta\left(\frac{k}{N}\right) \frac{k + \lambda_G}{\lambda_G} \mathcal{G}_k^{\lambda_G}(t), \quad t \in [-1, 1]. \tag{20}$$

It can be found in Dai & Xu (2013) (Chapter 4) that $L_N$ is near best, achieving the order of best approximation for $f \in W_p^r(\mathcal{S}^{d-1})$.

**Lemma D.1.2 (Lemma 1 in Fang et al. (2020))** *For $N \in \mathbb{N}$, $1 \le p \le \infty$, $r > 0$, and $f \in W_p^r(\mathcal{S}^{d-1})$, there holds*

$$\| f - L_N(f) \|_p \le C_\eta N^{-r} \cdot \| f \|_{W_\infty^r(\mathcal{S}^{d-1})}, \tag{21}$$

*where $C_\eta$ is a constant depending only on the function $\eta$ in defining $L_N$.*

Note that $\left( -\Delta_{\mathcal{S}^{d-1}} + \mathcal{I} \right)^{r/2}$ is a self-adjoint operator. For $\mathbf{x} \in \mathcal{S}^{d-1}$, recalling the definition of $L_N(f)$, we have

$$L_N(f)(\mathbf{x}) = \langle f, \ell_{N,d}(\langle \mathbf{x}, \cdot \rangle) \rangle_{\mathcal{L}_2(\mathcal{S}^{d-1})}$$

$$= \left\langle \left( -\Delta_{\mathcal{S}^{d-1}} + \mathcal{I} \right)^{r/2} f, \left( -\Delta_{\mathcal{S}^{d-1}} + \mathcal{I} \right)^{-r/2} \ell_{N,d}(\langle \mathbf{x}, \cdot \rangle) \right\rangle_{\mathcal{L}_2(\mathcal{S}^{d-1})}$$

$$= \int_{\mathcal{S}^{d-1}} F_r(\mathbf{y}) \cdot \xi_{N,r}(\langle \mathbf{x}, \mathbf{y} \rangle) \rho_\mathcal{X}(d\mathbf{y}). \tag{22}$$

Hereafter, we denote $F_r = \left( -\Delta_{\mathcal{S}^{d-1}} + \mathcal{I} \right)^{r/2} f$ and $\xi_{N,r}(\langle \mathbf{x}, \cdot \rangle) = \left( -\Delta_{\mathcal{S}^{d-1}} + \mathcal{I} \right)^{-r/2} \ell_{N,d}(\langle \mathbf{x}, \cdot \rangle)$. By the fractional power of the operator $\left( -\Delta_{\mathcal{S}^{d-1}} + \mathcal{I} \right)^{-r/2}$ in a distributional sense, $\xi_{N,r}(\cdot)$ is a polynomial of degree at most $2N$ written as:

$$\xi_{N,r}(t) = \sum_{k=0}^{2N} (1 + \lambda_k)^{-r/2} \eta\left(\frac{k}{N}\right) \frac{k + \lambda_G}{\lambda_G} \mathcal{G}_k^{\lambda_G}(t), \quad t \in [-1, 1]. \tag{23}$$

The fractional power of $(-\Delta_{\mathcal{S}^{d-1}} + \mathcal{I})$ caused by the regularity $f \in W_\infty^r(\mathcal{S}^{d-1})$ enables $r$-dependent error bound for discretizing $L_N(f)$: the larger the regularity $r$ becomes, the smaller the bound for approximation error gets.

Following Fang et al. (2020), the key idea for a constructing neural network that approximates $L_N(f)$ is to discretize the integral form (22) by $M$ random samples $\mathbf{y} = \{\mathbf{y_1}, \ldots, \mathbf{y_M}\}$ independently drawn from $\rho_{\mathcal{X}}$. We write the discretized version of (22) as :

$$\widehat{L}_{N,M}^{\boldsymbol{y}}(f)(\mathbf{x}) = \frac{1}{M} \sum_{i=1}^{M} F_r(\mathbf{y}_i) \cdot \xi_{N,r}(\langle \mathbf{x}, \mathbf{y}_i \rangle), \quad \forall \mathbf{x} \in \mathcal{S}^{d-1}. \tag{24}$$

Before estimating the distance between $L_N(f)$ and $\widehat{L}_{N,M}^{\boldsymbol{y}}(f)$, we need a Sobolev embedding property.

**Proposition D.1.3** *For $d \geq 5$, $1 \leq p \leq \infty$, and $s \geq \frac{3d-2}{4}$, the Sobolev space $W_p^s(\mathcal{S}^{d-1})$ is continuously embedded into $C(\mathcal{S}^{d-1})$, the space of continuous functions on $\mathcal{S}^{d-1}$, which implies*

$$\|f\|_\infty \leq c_0 \left( \frac{6}{\pi e} \right)^{\frac{d}{4}} \cdot \|f\|_{W_p^s(\mathcal{S}^{d-1})}, \qquad f \in W_p^s(\mathcal{S}^{d-1}),$$

*where $c_0$ is an absolute constant independent of $r$, $d$, $s$, and $f$.*

Proposition D.1.3 is motivated from Eq.(14) in Hesse (2006), where they proved $\|f\|_\infty \leq C_{s,d} \|f\|_{W_p^s(\mathcal{S}^{d-1})}, f \in W_p^s(\mathcal{S}^{d-1})$ for $s \geq \frac{d-1}{2}$. The constant obtained in Hesse (2006) is $C_{s,d} := \left( \frac{1}{\omega_d} \sum_{k=0}^\infty \frac{\mathcal{N}(k,d)}{(k+\frac{d-2}{2})^{2s}} \right)^{1/2}$, where $\omega_d$ is the surface of $d$-dimensional sphere. For large enough $d$, $(1/\omega_d)^{1/2}$ grows in the order of $\mathcal{O}\left( \left( \frac{d}{2\pi e} \right)^{d/4} \right)$. Then, by choosing $s \geq \frac{3d-2}{4}$, $(1/\omega_d)^{1/2}$ can be absorbed into the infinite sum making the constant $C_{s,d}$ converge in an asymptotic regime of $d$. It should be noted that the threshold on smoothness index (i.e., $s \geq \frac{3d-2}{4}$) is larger than that from Hesse (2006) (i.e., $s \geq \frac{d-1}{2}$), where they consider the fixed $d$. See Appendix E.1 for the proof. Next, we state the discretization lemma which provides a probabilistic bound on the difference $L_N(f) - \widehat{L}_{N,M}^{\boldsymbol{y}}(f)$.

**Lemma D.1.4** *Let $r \leq \frac{3d-2}{4}$ and $0 < \alpha < 1$. If $f \in W_\infty^r(\mathcal{S}^{d-1})$, then for any $M \in \mathbb{N}$ and $1 \leq N \leq d^\alpha + 1$, there exist $\mathbf{y} = \{y_1, y_2, \ldots, y_M\} \subset \mathcal{S}^{d-1}$ such that*

$$\left\| L_N(f) - \widehat{L}_{N,M}^{\boldsymbol{y}}(f) \right\|_\infty \leq \frac{6 \cdot C'' \left( \frac{6}{\pi e} \right)^{\frac{d}{4}} \|f\|_{W_\infty^r(\mathcal{S}^{d-1})} d^{N+\frac{3d-4r-2}{8}} (2N+1)^{\frac{3d-4r}{4}}}{\sqrt{M}},$$

*where $C'' > 0$ is a constant depending on $\alpha$ but independent of $r$, $f$, $N$, $M$, and $d$.*

Lemma D.1.4 is motivated by Lemma 2 in Fang et al. (2020). The main framework of the proof is based on the Sobolev embedding property in Proposition D.1.3 and the concentration inequality for random variables with values in a Hilbert space, which can be found in Smale & Zhou (2007). For the application of the concentration inequality, the random variable $\xi(\mathbf{y}_i) := F_r(\mathbf{y}_i) \xi_{N,r}(\langle \mathbf{x}, \mathbf{y}_i \rangle)$ in (24) needs to be bounded in $\| \cdot \|_{W_2^s(\mathcal{S}^{d-1})}$ norm for $s \geq \frac{3d-2}{4}$. See Appendix E.2 for the proof of the Lemma.

When compared with the technical proof of Lemma 2 from Fang et al. (2020), the most notable difference comes from tracking the explicit dependency on d in the constant factor. Specifically, under the fixed d setting, Fang et al. (2020) did not explicitly express how the constant $c_{s,r,d}$ (see the statement in their Lemma) depends on d. However, in our paper, since the main focus is how the approximation error behaves under $d \to \infty$, we need to keep tracking how d explicitly affects the bound. The result of Proposition 4.1 in our paper serves an important role in this tracking. Note that the constant $c_0$ is independent of $s, d, r, f$ in the bound of Proposition 4.1, and we obtain the bound decays at the rate $\left( \frac{6}{\pi e} \right)^{d/4}$. However, in Fang et al. (2020), they utilized the result from Hesse (2006); that is, $\|f\|_\infty \leq C_{s,d} \|f\|_{W_p^s(\mathbb{S}^{d-1})}, f \in W_p^s(\mathbb{S}^{d-1})$ for $s \geq \frac{d-1}{2}$. Here, note that the constant $C_{s,d}$ is a function of d, and since they work under the fixed d setting, they did not pay much attention to the dependency. Of course, since we are in an asymptotic setting, we use the Stirling's formula to see behaviors of $\mathcal{N}(k,d)$ as $d \to \infty$, whereas Fang et al. (2020) just used a simple calculation $\mathcal{N}(k,d) \leq c_d' k^{d-2}$, for some $c_d'$ dependent on d.

D.2 Construction of Deep ReLU Networks : Error bound for (III)

In this section, several useful tools for the construction of neural network for approximating function $\widehat{L}^{\boldsymbol{y}}_{N,M}(f)$ are introduced. Then, the full proof of our main theorem is presented.

The first key lemma is from Yarotsky (2017) wherein the neural network that approximates the quadratic function $x^2$ for $x \in [0, 1]$ is constructed.

**Lemma D.2.1 (Proposition 2 in Yarotsky (2017))** *For any positive integer $m \geq 1$, there exists a deep ReLU network*

$$\tilde{f}_m \in \mathcal{F}\big(m, \big(1, 5, \ldots, 5, 1\big)\big),$$

*such that $\tilde{f}_m \in [0, 1]$ and $\left|\tilde{f}_m(x) - x^2\right| \leq 2^{-2m-2}$, for all $x \in [0, 1]$.*

The main idea of Lemma D.2.1 is to approximate the quadratic function via $\tilde{f}_m(x) := x - \sum_{s=1}^{m} \frac{g_s(x)}{2^{2s}}$. Here, $g_s(x)$ is a $s$-compositions of sawtooth functions defined as

$$g(x) = 2\sigma(x) - 4\sigma(x - 1/2) + 2\sigma(x - 1).$$

Note that $g(x)$ can be implemented by a single layer ReLU network. Then, we can easily construct a ReLU network $\tilde{f}_m$, which belongs to $\mathcal{F}(m, (1, 5, \ldots, 5, 1))$.

Next lemma states that we can construct a neural network that can implement the multiplication operator.

**Lemma D.2.2** *For any positive integer $m \geq 1$, there exists a deep ReLU network*

$$Mult_m \in \mathcal{F}\big(m + 3, \big(2, 10, \ldots, 10, 1\big)\big),$$

*such that $Mult_m(x, y) \in [0, 1]$ and*

$$|Mult_m(x, y) - xy| \leq 2^{-2m-1},$$

*for all $x, y \in [0, 1]$. Moreover, $Mult_m(x, 0) = Mult_m(0, y) = 0$.*

The key idea for constructing $Mult_m(x, y)$ is to invoke the identity $xy = \frac{1}{4}((x + y)^2 - (x - y)^2)$. The first two hidden layers in the network are used to compute $|\frac{x+y}{2}| \in [0, 1]$ and $|\frac{x-y}{2}| \in [0, 1]$ via $|x| = \sigma(x) + \sigma(-x)$. Given the values $|\frac{x+y}{2}|$ and $|\frac{x-y}{2}|$ as inputs, $\tilde{f}_m$ in Lemma D.2.1 is used for approximating $\frac{1}{4}(x + y)^2$ and $\frac{1}{4}(x - y)^2$ in the identity. See Appendix E.3 for the detailed proof.

The final key ingredient is to construct a deep ReLU network that approximates univariate polynomial functions of degree $k \in \mathbb{N}$, that is $x^k$ for $x \in [0, 1]$.

**Lemma D.2.3** *For any positive integer $m \geq 1, N \geq 2$ and for $P = \lceil \log_2(N) \rceil$, there exists a deep ReLU network*

$$Poly_m^{\{N\}} \in \mathcal{F}\big(L, \big(1, 11N, \ldots, 11N, 2^P\big), \mathcal{N}\big),$$

*with the depth $L = m + (m+4)\big(\lceil \log_2(N) \rceil - 1\big)$ and the number of parameters $\mathcal{N} \leq 202N \cdot (m+3)$ such that $Poly_m^{\{N\}}(x) \in [0, 1]^{2^P}$ and*

$$\left|Poly_m^j(x) - x^j\right| \leq P^2 \cdot 2^{-2m-1} \quad \text{for all} \quad j \in \{1, \ldots, 2^p\}$$

*for all $x \in [0, 1]$.*

Note that the network $Poly_m^{\{N\}}(x) := \{Poly_m^1(x), \ldots, Poly_m^{2^P}(x)\}$ with $P = \lceil \log_2(N) \rceil$ provides approximations to monomials $x^j$ of degree up to $2N$ for $x \in [0, 1]$ at its final output.

The key idea for the construction is to employ a tree structure; that is, the width of the network at $((m+1) + (m+4) \cdot j)^{\text{th}}$ hidden layer is doubled from that at $((m+1) + (m+4) \cdot (j-1))^{\text{th}}$ hidden

layer for $j \in \{1, \ldots, p-1\}$ as

$$\underbrace{\left\{\mathrm{Poly}_m^1(x), \ldots, \mathrm{Poly}_m^{2^{j-1}}(x)\right\}}_{((m+1)+(m+4)\cdot(j-1))^{\mathrm{th}}\mathrm{layer}}$$

$$\rightarrow \underbrace{\left\{\mathrm{Poly}_m^1(x), \ldots, \mathrm{Poly}_m^{2^{j-1}}(x), \mathrm{Mult}_m\big(x, \mathrm{Poly}_m^{2^{j-1}}(x)\big), \ldots, \tilde{f}_m\big(\mathrm{Poly}_m^{2^{j-1}}(x)\big)\right\}}_{((m+1)+(m+4)\cdot j)^{\mathrm{th}} \mathrm{layer}}. \quad (25)$$

The first $2^{j-1}$ input values in $((m+1)+(m+4)\cdot j)^{\mathrm{th}}$ hidden layer is exactly copied from input values at the $((m+1)+(m+4)\cdot(j-1))^{\mathrm{th}}$ hidden layer. The remaining $2^{j-1}$ input values in $((m+1)+(m+4)\cdot j)^{\mathrm{th}}$ hidden layer approximates monomials $\{x^{2^{j-1}+1}, \ldots, x^{2^j}\}$ through $\tilde{f}_m$ and $\mathrm{Mult}_m$ operations in Lemmas D.2.1 and D.2.2. The approximation error can be obtained via proof by induction. Readers can find the detailed proof in the Appendix E.4 with the exact descriptions on the construction of $\mathrm{Poly}_m^{\{\mathrm{N}\}}$.

Finally, we are ready to state Proposition D.2.4 on the construction of network $\tilde{f}$ which approximates $\widehat{L}_{N,M}^{\boldsymbol{y}}(f)$.

**Proposition D.2.4** *Let $0 < \alpha < 1, m, N, M \in \mathbb{N}$ with $1 \leq N \leq d^\alpha + 1$. For any function $f \in W_\infty^r(\mathcal{S}^{d-1})$ with $r > 0$, define $\widehat{L}_{N,M}^{\boldsymbol{y}}(f)$ in (24). Then, there exists a network*

$$\tilde{f} \in \mathcal{F}\big(L, \big(d, 22NM, \ldots, 22NM, 1\big), \mathcal{N}\big)$$

*with depth $L = (m+4)\lceil\log_2(2N)\rceil$ and number of parameters $\mathcal{N} \leq M(2d + 404N \cdot (m+3) + 2N + 4) + 1$ such that*

$$\left\|\widehat{L}_{N,M}^{\boldsymbol{y}}(f) - \tilde{f}\right\|_\infty \leq C_\eta' \cdot \|f\|_{W_\infty^r(\mathcal{S}^{d-1})} \, d^{2N} \big(\log_2(2N)\big)^2 2^{-2m}, \quad (26)$$

*where $C_\eta'$ is a positive constant depending on $\eta$ and $\alpha$, but not on $d, r, m, N, M$ or $f$.*

A detailed proof for Proposition D.2.4 is deferred in the Appendix E.5.

Given the input data $\mathbf{x} \in \mathcal{S}^{d-1}$, recall the definition of $\widehat{L}_{N,M}^{\boldsymbol{y}}(f)(\mathbf{x})$ in (24). The crux of the whole construction procedure of our network is to build the sub-network which approximates $\xi_{N,r}(\langle\mathbf{x}, \mathbf{y}_i\rangle)$ for each $i \in [M]$. The key observation is that $\xi_{N,r}(\langle\mathbf{x}, \mathbf{y}_i\rangle)$ is the weighted sum of univariate polynomials of degree up to $2N$. Let $u_i = \langle\mathbf{x}, \mathbf{y}_i\rangle$. With the properly defined constant $\alpha_{i,q}$ (see its definition in the Appendix E.5), $\xi_{N,r}(\langle\mathbf{x}, \mathbf{y}_i\rangle)$ can be re-written as $\xi_{N,r}(u_i) := \sum_{q=0}^{2N} \alpha_{i,q}|u_i|^q$. Since $|u_i| \in [0, 1]$, with the help of network constructed in Lemma D.2.3 with $P = \lceil\log_2(2N)\rceil$, the sub-network that approximates $\xi_{N,r}(u_i)$ is easily constructed. Recall this is enabled through the reproducing property of the kernel of $\mathcal{H}_k^d$ for $0 \leq K \leq 2N$.

# E    PROOFS OF STATEMENTS IN APPENDIX B AND COROLLARY 3.3

## E.1    PROOF OF PROPOSITION D.1.3

**Proposition D.1.3** *For $d \geq 5$, $1 \leq p \leq \infty$, and $s \geq \frac{3d-2}{4}$, the Sobolev space $W_p^s(\mathcal{S}^{d-1})$ is continuously embedded into $C(\mathcal{S}^{d-1})$, the space of continuous functions on $\mathcal{S}^{d-1}$, which implies*

$$\|f\|_\infty \leq c_0\left(\frac{6}{\pi e}\right)^{\frac{d}{4}} \cdot \|f\|_{W_p^s(\mathcal{S}^{d-1})}, \qquad f \in W_p^s(\mathcal{S}^{d-1}),$$

*where $c_0$ is an absolute constant independent of $r$, $d$, $s$, and $f$.*

*Proof.* For $f \in W_p^s(\mathcal{S}^{d-1})$, by Sobolev embedding Lemma (see Hesse (2006) Eq. 14, p. 420), the infinity norm can be bounded by the Sobolev norm as

$$\|f\|_\infty \leq C_{s,d} \cdot \|f\|_{W_2^s(\mathcal{S}^{d-1})}, \quad (27)$$

where the constant $C_{s,d}$ is defined with its square as

$$C_{s,d}^2 := \frac{1}{\omega_d} \sum_{k=0}^{\infty} \frac{\mathcal{N}(k,d)}{(k + \frac{d-2}{2})^{2s}} \tag{28}$$

with $\omega_d = 2\pi^{\frac{d}{2}}/\Gamma(\frac{d}{2})$. Recalling (3), it is easy to see that by Stirling's formula, for large $d$, $\mathcal{N}(k,d) = (k + \frac{d-2}{2})^{d-2}(1 + \mathcal{O}(\frac{1}{d}))$. Also, we have

$$\Gamma\left(\frac{d}{2}\right) = \frac{2}{d}\Gamma\left(\frac{d}{2} + 1\right) = 2\sqrt{\frac{\pi}{d}}\left(\frac{d}{2e}\right)^{\frac{d}{2}}\left(1 + \mathcal{O}\left(\frac{1}{d}\right)\right). \tag{29}$$

When $s > \frac{d-1}{2}$, we have

$$\sum_{k=0}^{\infty}\left(k + \frac{d-2}{2}\right)^{d-2-2s} \le \int_{\frac{d-2}{2}-1}^{\infty} t^{d-2-2s}dt = \frac{1}{2s+1-d}\left(\frac{d-2}{2} - 1\right)^{d-1-2s}.$$

Observe that $d \ge 5$, we have $\frac{d-2}{2} - 1 \ge \frac{d}{12}$. Thus, when $s \ge \frac{3d-2}{4}$, we have $2s + 1 - d \ge d/2$ and thereby (28) is bounded as

$$C_{s,d}^2 \le \sqrt{\frac{\pi}{d}}\left(\frac{d}{2\pi e}\right)^{\frac{d}{2}}\frac{2}{d}\left(\frac{d}{12}\right)^{-\frac{d}{2}}\left(1 + \mathcal{O}\left(\frac{1}{d}\right)\right) = \frac{2\sqrt{\pi}}{d\sqrt{d}}\left(\frac{6}{\pi e}\right)^{\frac{d}{2}}\left(1 + \mathcal{O}\left(\frac{1}{d}\right)\right).$$

Then, there exists an absolute constant $c_0$ such that

$$C_{s,d}^2 \le c_0^2 \left(\frac{6}{\pi e}\right)^{\frac{d}{2}}, \qquad \forall d \ge 5.$$

This yields the claim. $\qquad\square$

### E.2 PROOF OF LEMMA D.1.4

**Lemma D.1.4** *Let $0 < r \le \frac{3d-2}{4}$ and $0 < \alpha < 1$. If $f \in W_{\infty}^r(\mathcal{S}^{d-1})$, then for any $M \in \mathbb{N}$ and $1 \le N \le d^{\alpha} + 1$, there exist $\mathbf{y} = \{y_1, y_2, \ldots, y_M\} \subset \mathcal{S}^{d-1}$ such that*

$$\left\| L_N(f) - \widehat{L}_{N,M}^{\mathbf{y}}(f) \right\|_{\infty} \le \frac{6 \cdot C''\left(\frac{6}{\pi e}\right)^{\frac{d}{4}} \|f\|_{W_{\infty}^r(\mathcal{S}^{d-1})} d^{N + \frac{3d-4r-2}{8}}(2N+1)^{\frac{3d-4r}{4}}}{\sqrt{M}},$$

*where $C'' > 0$ is a constant depending on $\alpha$ but independent of $r$, $f$, $N$, $M$, and $d$.*

*Proof.* We recall the following probability inequality for random variables with values in a Hilbert space which can be found in Smale & Zhou (2007).

**Lemma E.2.1** *Let $(H, \|\cdot\|)$ be a Hilbert space and $\xi$ be a random variable on $(Y, \rho_{\mathcal{X}})$ with values in $H$. Assume $\|\xi\| \le \mathcal{M} < \infty$ almost surely. Denote $\sigma^2(\xi) = \mathbb{E}(\|\xi\|^2)$. Let $\{y_i\}_{i=1}^M$ be independent samples from $\rho_{\mathcal{X}}$. Then for any $0 < \delta < 1$, we have with probability at least $1 - \delta$,*

$$\left\| \frac{1}{M}\sum_{i=1}^M \xi(y_i) - \mathbb{E}(\xi) \right\|_H \le \frac{2\mathcal{M}\log\left(\frac{2}{\delta}\right)}{M} + \sqrt{\frac{2\sigma^2(\xi)\log\left(\frac{2}{\delta}\right)}{M}}. \tag{30}$$

Let us define the random variable $\xi$ on $(\mathcal{S}^{d-1}, \rho_{\mathcal{X}})$ with values in $H$ given by

$$\xi(y) = F_r(y)\sum_{k=0}^{2N}(1 + \lambda_k)^{-r/2}\eta\left(\frac{k}{N}\right)Z_k(y, \cdot), \qquad y \in \mathcal{S}^{d-1}. \tag{31}$$

To bound the norm $\|\xi\| = \|\xi(y)\|_{W_2^s}^2$, we set $s = \frac{3d-2}{4}$ and recall the norm of $W_2^s(\mathcal{S}^{d-1})$ given with $p = 2$ and for $y \in \mathcal{S}^{d-1}$,

$$\|\xi(y)\|_{W_2^s(\mathcal{S}^{d-1})} = \left\| F_r(y)\sum_{k=0}^{2N}(1 + \lambda_k)^{\frac{s-r}{2}}\eta\left(\frac{k}{N}\right)Z_k(y, \cdot)\right\|_{L_2(\mathcal{S}^{d-1})}. \tag{32}$$

Recall $\lambda_k = k(k + d - 2)$. Then, for $0 \leq k \leq 2N$, $d \geq 3$, we have $k^2 < 1 + \lambda_k \leq dk^2$. We find $(1 + \lambda_k)^{s-r} \leq d^{s-r}k^{2(s-r)}$ by $s = \frac{3d-2}{4} \geq r$ $(\because s - r \geq 0)$. Also note that $0 \leq \eta(t) \leq 1$ for $t \in [0, 2]$. Employing Stirling's formula $d! = \sqrt{2\pi d}\left(\frac{d}{e}\right)^d\left(1 + \mathcal{O}(1/d)\right)$ in the expression (3) for $\mathcal{N}(k, d)$ yields $\mathcal{N}(k, d) \leq Cd^k$ for $0 \leq k \leq 2N$ and some constant $C$ depending on $\alpha$ but independent of $d$. By using the identity $Z_k(y, y) = \mathcal{N}(k, d)$ (see Corollary 1.2.7. in Dai & Xu (2013)), $\|\xi\|_{W_2^s(\mathcal{S}^{d-1})}^2$ can be bounded as

$$F_r(y)^2 \cdot \sum_{k=0}^{2N}(1 + \lambda_k)^{s-r}\eta^2\left(\frac{k}{N}\right)\mathcal{N}(k, d) = F_r(y)^2 \cdot \left(1 + \sum_{k=1}^{2N}(1 + \lambda_k)^{s-r}\eta^2\left(\frac{k}{N}\right)\mathcal{N}(k, d)\right)$$

$$\leq F_r(y)^2 \cdot \left(1 + C \cdot d^{s-r} \cdot \sum_{k=1}^{2N}k^{2(s-r)}d^k\right)$$

$$\leq F_r(y)^2 \cdot \left(1 + C \cdot d^{2N+s-r} \cdot \sum_{k=1}^{2N}k^{2(s-r)}\right),$$

while the term $\sum_{k=1}^{2N} k^{2(s-r)}$ with $s - r \geq 0$ can be bounded as

$$\sum_{k=1}^{2N}k^{2(s-r)} \leq \int_1^{2N+1}x^{2(s-r)}dx \leq \frac{1}{2(s-r)+1}(2N+1)^{2(s-r)+1}.$$

Combining this with the definitions of the norm $\|f\|_{W_\infty^r(\mathcal{S}^{d-1})}$, we know that $\|\xi(y)\|_{W_2^s}^2$ can be bounded as

$$\|\xi(y)\|_{W_2^s}^2 \leq C'^2\|f\|_{W_\infty^r(\mathcal{S}^{d-1})}^2 \cdot d^{2N+s-r}(2N+1)^{2(s-r)+1},$$

where $C'$ is a constant depending on $\alpha$ but independent of $r$, $s$, $f$, $N$, and $d$. Thus the random variable $\xi$ satisfies the condition $\|\xi\| \leq \mathcal{M} < \infty$ in Lemma E.2.1 with $\mathcal{M} = C'\|f\|_{W_\infty^r(\mathcal{S}^{d-1})}d^{N+\frac{s-r}{2}}(2N+1)^{(s-r)+\frac{1}{2}}$. So by Lemma E.2.1, with $\delta = \frac{1}{2}$ and $\sigma^2(\xi) \leq \mathcal{M}^2$, we know from the positive measure of the sample set that there exists a set of points $\mathbf{y} = \{y_i\}_{i=1}^M \in \mathcal{S}^{d-1}$ such that

$$\left\|\frac{1}{M}\sum_{i=1}^M\xi(y_i) - \mathbb{E}(\xi)\right\|_{\mathrm{H}} = \left\|L_N(f) - \widehat{L}_{N,M}^{\mathbf{y}}(f)\right\|_{W_2^s(\mathcal{S}^{d-1})}$$

$$\leq \frac{6 \cdot C'\|f\|_{W_\infty^r(\mathcal{S}^{d-1})}d^{N+\frac{s-r}{2}}(2N+1)^{(s-r)+\frac{1}{2}}}{\sqrt{M}}. \tag{33}$$

Since $s = \frac{3d-2}{4}$, combining the result from Proposition D.1.3 with (33) yields

$$\left\|L_N(f) - \widehat{L}_{N,M}^{\mathbf{y}}(f)\right\|_\infty \leq \frac{6 \cdot C''\left(\frac{6}{\pi e}\right)^{\frac{d}{4}}\|f\|_{W_\infty^r(\mathcal{S}^{d-1})}d^{N+\frac{3d-4r-2}{8}}(2N+1)^{\frac{3d-4r}{4}}}{\sqrt{M}},$$

where $C'' > 0$ is a constant depending on $\alpha$ but independent of $r$, $f$, $N$, $M$, and $d$. $\qquad\square$

### E.3 PROOF OF LEMMA D.2.2

**Lemma D.2.2** *For any positive integer $m \geq 1$, there exists a deep ReLU network*

$$Mult_m \in \mathcal{F}\big(m + 3, (2, 10, \ldots, 10, 1)\big),$$

*such that $Mult_m(x, y) \in [0, 1]$ and*

$$|Mult_m(x, y) - xy| \leq 2^{-2m-1},$$

*for all $x, y \in [0, 1]$. Moreover, $Mult_m(x, 0) = Mult_m(0, y) = 0$.*

*Proof.* Given input $(x, y)$, the network $\text{Mult}_m(x, y)$ computes in the first hidden layer

$$(x, y) \to \left\{ \sigma\left(\frac{x + y}{2}\right), \sigma\left(-\left(\frac{x + y}{2}\right)\right), \sigma\left(\frac{x - y}{2}\right), \sigma\left(-\left(\frac{x - y}{2}\right)\right) \right\}.$$

By using the equality $|x| = \sigma(x) + \sigma(-x)$ for $x \in [0, 1]$, the network computes in the second hidden layer

$$(x, y) \to \left\{ \sigma\left(\left|\frac{x + y}{2}\right|\right), \sigma\left(\left|\frac{x - y}{2}\right|\right) \right\}.$$

Note $\sigma\left(\left|\frac{x+y}{2}\right|\right), \sigma\left(\left|\frac{x-y}{2}\right|\right) \in [0, 1]$, and $\sigma\left(\left|\frac{x+y}{2}\right|\right) = \left|\frac{x+y}{2}\right|, \sigma\left(\left|\frac{x-y}{2}\right|\right) = \left|\frac{x-y}{2}\right|$. We apply the network $\tilde{f}_m$ on the two components respectively. This gives a network of $(m + 2)$ hidden layers with width vector $(2, 10, \ldots, 10, 2)$ that computes

$$(x, y) \to \left\{ \sigma\left(\tilde{f}_m\left(\left|\frac{x + y}{2}\right|\right)\right), \sigma\left(\tilde{f}_m\left(\left|\frac{x - y}{2}\right|\right)\right) \right\}. \tag{34}$$

The network $\text{Mult}_m$ computes (34) in the $(m + 3)^{\text{th}}$ hidden layer. Since $\tilde{f}_m \in [0, 1]$, $\sigma\left(\tilde{f}_m(x)\right) = \tilde{f}_m(x)$. In the output layer, the network value is computed as

$$\text{Mult}_m(x, y) := \tilde{f}_m\left(\left|\frac{x + y}{2}\right|\right) - \tilde{f}_m\left(\left|\frac{x - y}{2}\right|\right). \tag{35}$$

Since $\tilde{f}_m$ is an increasing function in argument, $\text{Mult}_m(x, y) \geq 0$, and since $\tilde{f}_m \in [0, 1]$, $\text{Mult}_m(x, y) \leq 1$. By identity, $xy = \left|\frac{x+y}{2}\right|^2 - \left|\frac{x-y}{2}\right|^2$, and Lemma D.2.1, the error is computed as follows:

$$\left|\text{Mult}_m(x, y) - xy\right| \leq \left|\tilde{f}_m\left(\left|\frac{x + y}{2}\right|\right) - \left(\left|\frac{x + y}{2}\right|\right)^2\right| + \left|\tilde{f}_m\left(\left|\frac{x - y}{2}\right|\right) - \left(\left|\frac{x - y}{2}\right|\right)^2\right|$$
$$\leq 2^{-2m-1}.$$

If either $x = 0$ or $y = 0$, by the definition of (35), we have $\text{Mult}_m(x, 0) = \text{Mult}_m(0, y) = 0$. $\square$

### E.4  Proof of Lemma D.2.3

**Lemma D.2.3** *For any positive integer $m \geq 1, N \geq 2$ and for $P = \lceil \log_2(N) \rceil$, there exists a deep ReLU network*

$$Poly_m^{\{N\}} \in \mathcal{F}\left(L, \left(1, 11N, \ldots, 11N, 2^P\right), \mathcal{N}\right),$$

*with the depth $L = m + (m+4)\left(\lceil \log_2(N) \rceil - 1\right)$ and the number of parameters $\mathcal{N} \leq 202N \cdot (m+3)$ such that $Poly_m^{\{N\}}(x) \in [0, 1]^{2^P}$ and*

$$\left|Poly_m^j(x) - x^j\right| \leq P^2 \cdot 2^{-2m-1} \quad \text{for all} \quad j \in \{1, \ldots, 2^P\}$$

*for all $x \in [0, 1]$.*

*Proof.* Let us describe the construction of the network $Poly_m^{\{N\}}$. With the application of Lemma D.2.1, in the $(m + 1)^{\text{th}}$ hidden layer, the network computes

$$x \to \left\{ \sigma(x), \sigma(\tilde{f}_m(x)) \right\}$$

with the width $\mathbf{p} = (1, 5, \ldots, 5, 2)$. For approximating $x^3$, the network $\text{Mult}_m$ is applied on the pair $(\sigma(x), \sigma(\tilde{f}_m(x)))$, and for approximating $x^4$, the network $\tilde{f}_m$ is applied on the $\sigma(\tilde{f}_m(x))$. Therefore, in the $\{(m + 1) + (m + 4)\}^{\text{th}}$ hidden layer, the network $Poly_m^{\{N\}}$ computes

$$x \to \left\{ \sigma(x), \sigma(\tilde{f}_m(x)), \sigma\left(\text{Mult}_m(x, \tilde{f}_m(x))\right), \sigma\left(\tilde{f}_m(\tilde{f}_m(x))\right) \right\}. \tag{36}$$

Note that each component in the hidden layer is in $[0, 1]$ by Lemmas D.2.1 and D.2.2. This procedure is continued until a following vector is in the final output layer,

$$x \rightarrow \left\{ \mathrm{Poly}_m^1(x), \ldots, \mathrm{Poly}_m^{2^{P-1}}(x), \mathrm{Mult}_m(x, \mathrm{Poly}_m^{2^{P-1}}(x)), \ldots, \tilde{f}_m\big(\tilde{f}_m \ldots (\tilde{f}_m(x))\big) \right\} \in [0, 1]^{2^P}.$$

The resulting network is referred as $\mathrm{Poly}_m^{\{N\}}$ and has $m + (m+4)\big(\lceil \log_2(N) \rceil - 1\big)$ hidden layers. Recall $P = \lceil \log_2(N) \rceil$. By the construction procedure of the network, we can compute the upper bound of maximum width as,

$$2^{\lceil \log_2(N) \rceil - 1} + \left\{ 10 \cdot \left( 2^{\lceil \log_2(N) \rceil - 1} - 1 \right) + 5 \right\} \le 11 \cdot 2^{\lceil \log_2(N) \rceil - 1} \le 11N, \qquad (37)$$

where we use $\lceil \log_2(N) \rceil \le \log_2(N) + 1$ in the second inequality. Now, we need to count the number of active parameters in the network. For $k \in \{1, \ldots, \lceil \log_2(N) \rceil\}$, we compute the upper bound on the total number of active parameters in-between following hidden layers:

$$\left\{ \mathrm{Poly}_m^1(x), \ldots, \mathrm{Poly}_m^{2^{k-1}}(x) \right\} \rightarrow \left\{ \mathrm{Poly}_m^1(x), \ldots, \mathrm{Poly}_m^{2^{k-1}}(x), \mathrm{Poly}_m^{2^{k-1}+1}(x), \ldots, \mathrm{Poly}_m^{2^k}(x) \right\}.$$
$$(38)$$

Think of a network that takes the hidden layer in the left hand side of (38) as an input, and gives the hidden layer in the right hand side of (38) as an output. It is easy to count the number of active parameters in input, hidden, and output layers, separately as follows:

$$\begin{cases} \text{Input layer} & : 2^{k-1} + 1 + 2 \cdot \left( 2^{k-1} - 1 \right) = 3 \cdot 2^{k-1} - 1. \\ \text{Hidden layers} & : (m+2) \cdot 2^{k-1} + 100 \cdot (m+2) \cdot (2^{k-1} - 1) + 25 \cdot (m+2) = (m+2)(101 \cdot 2^{k-1} - 75). \\ \text{Output layer} & : 2^{k-1} + 10 \cdot (2^{k-1} - 1) + 5 = 11 \cdot 2^{k-1} - 5. \end{cases}$$

Since the $k$ runs over $\{1, \ldots, \lceil \log_2(N) \rceil\}$, the total number of active parameters can be bounded as:

$$\sum_{k=1}^{\lceil \log_2(N) \rceil} \left\{ \left( m + 2 \right)\left( 101 \cdot 2^{k-1} - 75 \right) + \left( 14 \cdot 2^{k-1} - 6 \right) \right\}$$
$$\le (m+2) \cdot 101 \sum_{k=1}^{\lceil \log_2(N) \rceil} 2^{k-1} + 14 \cdot \sum_{k=1}^{\lceil \log_2(N) \rceil} 2^{k-1}$$
$$\le 202N \cdot (m+3).$$

The approximation error is proved via induction on the number of iterated multiplications $P = \lceil \log_2(N) \rceil$. For $P = 1$, that is $N = 2$, we have

$$\left| x^2 - \tilde{f}_m(x) \right| \le 2^{-2m-1}$$

by Lemma D.2.1. For the convenience of notation, denote $\tilde{x}^a := \mathrm{Poly}_m^a(x)$ for some positive integer $a$. For $P = k - 1$, assume a following holds

$$\left| x^j - \tilde{x}^j \right| \le 3^{k-2} \cdot 2^{-2m-1} \quad \text{for} \quad j \in \{1, \ldots, 2^{k-1}\}.$$

Then, for $P = k$, we want to prove

$$\left| x^j - \tilde{x}^j \right| \le 3^{k-1} \cdot 2^{-2m-1} \quad \text{for} \quad j \in \{1, \ldots, 2^k\}.$$

By the construction of neural network and induction assumption, for $j \in \{1, \ldots, 2^{k-1}\}$, we have $\left| x^j - \tilde{x}^j \right| \le 3^{k-2} \cdot 2^{-2m-1} \le 3^{k-1} \cdot 2^{-2m-1}$. For any $j \in \{2^{k-1} + 1, \ldots, 2^k\}$, find any $a, b \in \{1, \ldots, 2^{k-1}\}$ such that $j = a + b$. Then, for $x \in [0, 1]$,

$$\left| x^{a+b} - \mathrm{Mult}_m\big(\tilde{x}^a, \tilde{x}^b\big) \right| \le \left| x^{a+b} - \tilde{x}^a \cdot \tilde{x}^b \right| + \left| \tilde{x}^a \cdot \tilde{x}^b - \mathrm{Mult}_m\big(\tilde{x}^a, \tilde{x}^b\big) \right|$$
$$\le x^a \left| x^b - \tilde{x}^b \right| + \tilde{x}^b \left| x^a - \tilde{x}^a \right| + \left| \tilde{x}^a \cdot \tilde{x}^b - \mathrm{Mult}_m\big(\tilde{x}^a, \tilde{x}^b\big) \right|$$
$$\le 3^{k-2} \cdot 2^{-2m-1} + 3^{k-2} \cdot 2^{-2m-1} + 2^{-2m-1} \le 3^{k-1} \cdot 2^{-2m-1}.$$

By using the fact $\log_2(3) < 2$, we can deduce $3^{k-1} < P^2$ and conclude the proof. $\qquad \square$

### E.5  PROOF OF PROPOSITION D.2.4

**Proposition D.2.4** *Let* $0 < \alpha < 1, m, N, M \in \mathbb{N}$ *with* $1 \leq N \leq d^{\alpha} + 1$. *For any function* $f \in W_{\infty}^r(\mathcal{S}^{d-1})$ *with* $r > 0$, *define* $\widehat{L}_{N,M}^{\boldsymbol{y}}(f)$ *in* (24). *Then, there exists a network*

$$\tilde{f} \in \mathcal{F}\big(L, \big(d, 22NM, \dots, 22NM, 1\big), \mathcal{N}\big)$$

*with depth* $L = (m+4)\lceil \log_2(2N) \rceil$ *and number of parameters* $\mathcal{N} \leq M(2d + 404N \cdot (m+3) + 2N + 4) + 1$ *such that*

$$\left\| \widehat{L}_{N,M}^{\boldsymbol{y}}(f) - \tilde{f} \right\|_{\infty} \leq C_{\eta}^{'} \cdot \|f\|_{W_{\infty}^r(\mathcal{S}^{d-1})} \, d^{2N} \big(\log_2(2N)\big)^2 2^{-2m}, \tag{39}$$

*where* $C_{\eta}^{'}$ *is a positive constant depending on* $\eta$ *and* $\alpha$, *but not on* $d, r, m, N, M$ *or* $f$.

*Proof.* We adopt the shorthand notation denoting $[n] := \{1, 2, \dots, n\}$ and $[n]_0 := \{0, 1, \dots, n\}$ for $n \in \mathbb{N}$ in the proof.

Given the input data $\mathbf{x} \in \mathcal{S}^{d-1}$, recall the definition of $\widehat{L}_{N,M}^{\boldsymbol{y}}(f)(\mathbf{x})$ in (24). The crux of the whole construction procedure is to build the sub-network which approximates $\xi_{N,r}(\langle \mathbf{x}, \mathbf{y}_i \rangle)$ for each $i \in [M]$. First, observe that, by (4) and (23), $\xi_{N,r}(u_i)$ can be written as:

$$\xi_{N,r}(u_i) = \sum_{k=0}^{2N} (1 + \lambda_k)^{-\frac{r}{2}} \eta\left(\frac{k}{N}\right) \left\{ \frac{k + \lambda_{\mathrm{G}}}{\lambda_{\mathrm{G}}} \sum_{\ell=0}^{\lfloor \frac{k}{2} \rfloor} (-1)^{\ell} \frac{\Gamma\big(k - \ell + \lambda_{\mathrm{G}}\big)}{\Gamma(\lambda_{\mathrm{G}})\ell!(k - 2\ell)!} \big(2u_i\big)^{k-2\ell} \right\}, \tag{40}$$

for $i \in [M]$. The key observation is that Eq. (40) is the weighted sum of univariate polynomials of degree up to $2N$. We define a constant $c_{k,\ell,\eta,\lambda_k,r,d}$ as

$$c_{k,\ell,\eta,\lambda_k,r,d} := (1 + \lambda_k)^{-\frac{r}{2}} \eta\left(\frac{k}{N}\right) \frac{k + \lambda_{\mathrm{G}}}{\lambda_{\mathrm{G}}} \frac{(-1)^{\ell}\Gamma\big(k - \ell + \lambda_{\mathrm{G}}\big)2^{k-2\ell}}{\Gamma(\lambda_{\mathrm{G}})\ell!\big(k - 2\ell\big)!}. \tag{41}$$

For $i \in \{1, \dots, M\}$, set $\alpha_{i,q}$ as

$$\alpha_{i,q} = \begin{cases} \sum_{(k,\ell)\in\mathcal{A}_q} \big(-c_{k,\ell,\eta,\lambda_k,r,d}\big) & \text{if } u_i < 0 \text{ and } q \text{ is odd}, \\ \sum_{(k,\ell)\in\mathcal{A}_q} \big(c_{k,\ell,\eta,\lambda_k,r,d}\big) & \text{otherwise}, \end{cases} \tag{42}$$

where for each $q \in \{0, \dots, 2N\}$, the set $\mathcal{A}_q$ is given by $\mathcal{A}_q := \{(k, \ell) \in [2N]_0 \times [\lfloor k/2 \rfloor]_0 : k - 2\ell = q\}$. Then, (40) can be re-written as $\xi_{N,r}(u_i) := \sum_{q=0}^{2N} \alpha_{i,q}|u_i|^q$.

**1.  The Network Construction.** Now, we are ready for the construction of $\tilde{f}$. Through Lemma D.1.4, we know that there exists $\boldsymbol{y} = \{\mathbf{y}_1, \dots, \mathbf{y}_M\}$ that satisfies the bound (D.1.4). Then, for each $i \in [M]$, we put $\mathbf{y}_i \in \mathcal{S}^{d-1}$ as a weight vector that connects input $\mathbf{x}$ to the $(2i-1)^{\text{th}}$ and $(2i)^{\text{th}}$ nodes in the first hidden layer. Through this, $\tilde{f}$ computes in its first hidden layer $\mathbf{x} \to \{\sigma(\langle \mathbf{x}, \mathbf{y}_1 \rangle), \sigma(-\langle \mathbf{x}, \mathbf{y}_1 \rangle), \dots, \sigma(\langle \mathbf{x}, \mathbf{y}_M \rangle), \sigma(-\langle \mathbf{x}, \mathbf{y}_M \rangle)\} \in [0,1]^{2M}$. Then, by the identity $|x| = \sigma(x) + \sigma(-x)$ for $x \in \mathbb{R}$, the network computes in its second hidden layer $\mathbf{x} \to \{\sigma(|u_1|), \sigma(|u_2|), \dots, \sigma(|u_M|)\} \in [0,1]^M$, where $u_i := \langle \mathbf{x}, \mathbf{y}_i \rangle \in [-1,1]$ for $i \in [M]$. Since $\sigma(|u_i|) = |u_i| \in [0,1]$, $\text{Poly}_m^{\{2N\}}$ with $P = \lceil \log_2(2N) \rceil$ is applicable for each $\{|u_i|\}_{i=1}^M$, and it generates $\text{Poly}_m^q(|u_i|)$ with $q$ at most $4N$. Set $\mathcal{B}_{\max} := \max_{i=1,\dots,M} \left| \sum_{q=0}^{2N} \alpha_{i,q} \cdot \text{Poly}_m^q(|u_i|) \right|$.

Using the definition of the constant $\alpha_{i,q}$, the network $\tilde{f}$ computes in the $(m+4)\lceil \log_2(2N) \rceil$th hidden layer $\{\sigma(\sum_{q=0}^{2N} \alpha_{1,q}\text{Poly}_m^q(|u_1|) + 2\mathcal{B}_{\max}), \dots, \sigma(\sum_{q=0}^{2N} \alpha_{M,q}\text{Poly}_m^q(|u_M|) + 2\mathcal{B}_{\max})\} \in \mathbb{R}^M$. By the definition of $\mathcal{B}_{\max}$, it is easy to see each component in the hidden layer is positive. Set the weight of output layer as $\{\frac{1}{M}F_r(\mathbf{y}_i)\}_{i=1}^M$. Define $\mathcal{L}(|u_i|) := \sum_{q=0}^{2N} \alpha_{i,q} \cdot \text{Poly}_m^q(|u_i|) + 2 \cdot \mathcal{B}_{\max}$. Then, given the data $\boldsymbol{y} = \{\mathbf{y}_1, \dots, \mathbf{y}_M\}$, the network $\tilde{f}$ computes its final output as $\tilde{f}(\mathbf{x}) = \frac{1}{M}\sum_{j=1}^M F_r(\mathbf{y}_j) \cdot \big(\mathcal{L}(|\langle \mathbf{x}, \mathbf{y}_j \rangle|) - 2\mathcal{B}_{\max}\big) := \frac{1}{M}\sum_{i=1}^M F_r(\mathbf{y}_i) \cdot \mathcal{L}\big(\xi_{N,r}\big)\big(\langle \mathbf{x}, \mathbf{y}_i \rangle\big)$.

**2. The Width and Number of Active Parameters of $\tilde{f}$.** By the construction of network $\tilde{f}$ and the result of Lemma D.2.3, it is easy to see the maximum width of the network is $22NM$. Now, we

work on counting the number of active parameters in the network as

$$
\begin{cases}
\text{From Input to } 2^{\text{nd}} \text{ hidden layer} & : \quad 2Md + 2M. \\
\text{From } 2^{\text{nd}} \text{ to } \big((m+4)\lceil \log_2(2N)\rceil - 1\big)^{\text{th}} \text{ hidden layer} & : \quad 404NM \cdot (m+3). \\
\text{From } \big((m+4)\lceil \log_2(2N)\rceil - 1\big)^{\text{th}} \text{ hidden layer to output layer} & : \quad (2N+1)M + M + 1.
\end{cases}
$$

Summing up the total number yields the desired result.

**3. Approximation Error Computation.** A remaining thing is to calculate the approximation error:

$$
\left\| \widehat{L}_{N,M}^{\mathbf{y}}(f) - \tilde{f} \right\|_\infty = \sup_{\mathbf{x}\in\mathcal{S}^{d-1}} \left| \frac{1}{M} \sum_{i=1}^M F_r(\mathbf{y_i}) \cdot \xi_{N,r}(\langle \mathbf{x}, \mathbf{y_i}\rangle) - \frac{1}{M} \sum_{i=1}^M F_r(\mathbf{y_i}) \cdot \mathcal{L}\big(\xi_{N,r}\big)(\langle \mathbf{x}, \mathbf{y_i}\rangle) \right|
$$
$$
\leq \|f\|_{W_\infty^r(\mathcal{S}^{d-1})} \cdot \left\| \xi_{N,r} - \mathcal{L}\big(\xi_{N,r}\big) \right\|_\infty. \tag{43}
$$

Recall the definition of $\alpha_{i,q}$ in (42). Using Stirling's Formula, $\Gamma(n+1) = \sqrt{2\pi n}\big(\frac{n}{e}\big)^n (1+\mathcal{O}(1/n))$, we observe the behavior of Gegenbauer coefficient in (4) where $\lambda_{\mathrm{G}} = \frac{d-2}{2} \gg d^\alpha + 1 \geq N$, and find that it can be bounded as $C \cdot \lambda_{\mathrm{G}}^{k-\ell} \cdot 2^{k-2\ell}(1+\mathcal{O}(1/d))$, where $C > 0$ is a constant independent of $d$.

For $k \in \{0, 1, \dots, 2N\}$, combining the facts $(1+\lambda_k)^{-\frac{r}{2}} < 1$, $\eta(\cdot) \leq 1$, $\frac{k+\lambda_{\mathrm{G}}}{\lambda_{\mathrm{G}}} \leq 2$ for $k \leq 2N \leq 2(d^\alpha + 1)$ with $\lambda_{\mathrm{G}} = \frac{d-2}{2}$ yields

$$
\left| c_{k,\ell,\eta,\lambda_k,r,d} \right| \leq C'_\eta \cdot 2^{-\ell} \cdot d^{k-\ell}, \tag{44}
$$

where $C'_\eta > 0$ is a constant dependent on $\alpha$ and $\eta$. Recall $\mathcal{L}\big(\xi_{N,r}\big)(\langle \mathbf{x}, \mathbf{y}_j\rangle) := \sum_{q=0}^{2N} \alpha_{j,q} \cdot \mathrm{Poly}_m^q(|\langle \mathbf{x}, \mathbf{y}_j\rangle|)$ and note that $\sum_{q=0}^{2N} |\alpha_{j,q}| = \sum_{k=0}^{2N} \sum_{\ell=0}^{\lfloor \frac{k}{2} \rfloor} |c_{k,\ell,\eta,\lambda_k,r,d}|$. Then, we have

$$
\left\| \xi_{N,r} - \mathcal{L}\big(\xi_{N,r}\big) \right\|_\infty \leq \left( \sum_{k=0}^{2N} \sum_{\ell=0}^{\lfloor \frac{k}{2} \rfloor} |c_{k,\ell,\eta,\lambda_k,r,d}| \right) \cdot \left( \sup_{u\in[0,1]} \max_{q\in\{0,\dots,2N\}} |u^q - \mathrm{Poly}_m^q(u)| \right)
$$
$$
\leq C'_\eta \cdot \left( \sum_{k=0}^{2N} d^k \sum_{\ell=0}^{\lfloor \frac{k}{2} \rfloor} \frac{1}{(2d)^\ell} \right) \cdot \left( \big(\log_2(2N)\big)^2 \cdot 2^{-2m-1} \right)
$$

where we used the result from Lemma D.2.3 and (44) in the second inequality. Using $\sum_{\ell=0}^{\lfloor \frac{k}{2} \rfloor} \frac{1}{(2d)^\ell} \leq 2$ in the last inequality yields the claim. $\qquad\square$

### E.6 PROOF OF COROLLARY 3.3

**Corollary 3.3** *Let $0 < \alpha, \beta, \gamma < 1$ with $\gamma > \max\{\alpha, \beta\}$ and $N \in \mathbb{N}$ with $1 \leq N \leq d^\alpha + 1$. For any $f \in W_\infty^r(\mathcal{S}^{d-1})$ with $r > 0$, we have :*

*(I) For $\frac{3d-2}{4} - C_1 \leq r \leq \frac{3d-2}{4}$ with some constant $C_1 \geq 0$ independent of d, there exists a network*
$$
\tilde{f}^{(I)} \in \mathcal{F}\left(L, (d, 66N, 66N, \dots, 66N, 1), \mathcal{N}\right)
$$
*with depth $L = \mathcal{O}\left(d^\gamma \log_2 d\right)$ and the number of active parameters $\mathcal{N} = \mathcal{O}\left(d^{\max\{\alpha+\gamma, 1\}}\right)$, such that $\left\| f - \tilde{f}^{(I)} \right\|_\infty \leq C'_{\eta,\alpha,\beta,\gamma}\|f\|_{W_\infty^r(\mathcal{S}^{d-1})} d^{-d^\beta}$, where $C'_{\eta,\alpha,\beta,\gamma}$ is a constant depending only on $C_1, \eta, \alpha, \beta, \gamma$.*

*(II) For $r = \mathcal{O}(1)$ and $M = \mathcal{O}\left(9^d d^{\frac{9}{4}d}\right)$, there exists a network*
$$
\tilde{f}^{(II)} \in \mathcal{F}\left(L, \big(d, 22NM, \dots, 22NM, 1\big), \mathcal{N}\right)
$$
*with depth $L = \mathcal{O}\left(d^\gamma \log_2 d\right)$ and the number of active parameters $\mathcal{N} = \mathcal{O}\left(9^d d^{\frac{13}{4}d}\right)$ such that $\left\| f - \tilde{f}^{(II)} \right\|_\infty \leq C'_{\eta,\alpha,\beta,\gamma}\|f\|_{W_\infty^r(\mathcal{S}^{d-1})} d^{-\alpha r}$, where $C'_{\eta,\alpha,\beta,\gamma}$ is a constant depending only on $\eta, \alpha, \beta, \gamma$.*

*Proof.* By the results of Theorem 3.1, for $1 \leq N \leq d^{\alpha} + 1$, we have the following inequality on the approximation error

$$\left\| \tilde{f} - f \right\|_{\infty} \leq C_{\eta}^{''} \|f\|_{W_{\infty}^{r}(\mathcal{S}^{d-1})} \times$$

$$\max \left\{ N^{-r}, \frac{\left(\frac{6}{\pi e}\right)^{\frac{d}{4}} d^{N + \frac{3d-4r-2}{8}} (2N+1)^{\frac{3d-4r}{4}}}{\sqrt{M}}, d^{2N} \left( \log_2(2N) \right)^2 2^{-2m} \right\}, \quad (45)$$

where $C_{\eta}^{''}$ is a constant dependent on $\eta$, and independent on $d, r, N, M$ or $f$. We divide the proof into two cases.

*(I) $r = \mathcal{O}(d)$ and any integer $M \geq 1$*

For the first term in (45), since $N = \lceil d^{\alpha} \rceil$, we know that $N^{-r} = \mathcal{O}(d^{-\alpha r}) = \mathcal{O}(d^{-d^{\beta}})$ with any $0 < \beta < 1$. This is due to the assumption that $\frac{3d-2}{4} - C_1 \leq r \leq \frac{3d-2}{4}$, which implies $d = \mathcal{O}(r)$ and $d^{\beta} = o(r)$.

For the second term in (45), since $N = \lceil d^{\alpha} \rceil$ with $0 < \alpha < 1$, we know that it is bounded by

$$\frac{\left(\frac{6}{\pi e}\right)^{\frac{d}{4}} d^{N + \frac{3d-4r-2}{8}} (2N+1)^{\frac{3d-4r}{4}}}{\sqrt{M}} \leq \frac{\left(\frac{6}{\pi e}\right)^{\frac{d}{4}} d^{d^{\alpha} + \frac{3d-4r+6}{8}} (3d^{\alpha})^{\frac{3d-4r}{4}}}{\sqrt{M}}. \quad (46)$$

As $\frac{3d-2}{4} - C_1 \leq r \leq \frac{3d-2}{4}$, we know the term on the right hand side of (46) can be written as $\left(\frac{6}{\pi e}\right)^{\frac{d}{4}} d^{d^{\alpha} + \mathcal{O}(1)} 3^{\mathcal{O}(1)} / \sqrt{M}$. To show that the bound is of order $\mathcal{O}(d^{-d^{\beta}})$, we multiply the bound by $d^{d^{\beta}}$, take the logarithm, and find that for any $0 < \alpha, \beta < 1$,

$$\log \left( \left(\frac{6}{\pi e}\right)^{\frac{d}{4}} d^{d^{\alpha} + d^{\beta} + \mathcal{O}(1)} \right) \leq \frac{d}{4} \log \left(\frac{6}{\pi e}\right) + \left(d^{\alpha} + d^{\beta} + \mathcal{O}(1)\right) \log(d) \to -\infty,$$

as $d \to \infty$. Hence, there exists a constant $C_{\alpha,\beta} > 0$ depending only on $C_1, \alpha, \beta$ such that

$$\frac{\left(\frac{6}{\pi e}\right)^{\frac{d}{4}} d^{N + \frac{3d-4r-2}{8}} (2N+1)^{\frac{3d-4r}{4}}}{\sqrt{M}} \leq C_{\alpha,\beta} d^{-d^{\beta}},$$

for any fixed $M \in \mathbb{N}$. In our proof, we simply choose $M = 3$. For the third term in (45), take $m = \lceil d^{\gamma} \rceil$ with $\max\{\alpha, \beta\} < \gamma < 1$, then there exists a constant $C_{\alpha,\beta,\gamma}$ depending on $\alpha, \beta, \gamma$ such that

$$d^{2N} \left( \log_2(2N) \right)^2 2^{-2m} < d^{2d^{\alpha}+2} \left( 2 + \log_2(d) \right)^2 2^{-2m} \leq d^{3d^{\alpha}} 2^{-2m} \leq C_{\alpha,\beta,\gamma} d^{-d^{\beta}},$$

where $\log_2(2d^{\alpha} + 2) \leq \log_2(4d^{\alpha}) < 2 + \log_2(d)$ is used in the first inequality, and the last inequality follows from the same argument as above, of multiplying with $d^{d^{\beta}}$ and taking the logarithm. Combining all the analyses above, we have

$$\left\| \tilde{f} - f \right\|_{\infty} \leq C_{\eta,\alpha,\beta,\gamma}^{'} \|f\|_{W_{\infty}^{r}(\mathcal{S}^{d-1})} d^{-d^{\beta}},$$

where $C_{\eta,\alpha,\beta,\gamma}^{'} > 0$ is a constant dependent on $\eta, \alpha, \beta, \gamma$, and $C_1$.

Recall from Proposition D.2.4, $\tilde{f}$ is a network with depth $L = (m+4) \lceil \log_2(2N) \rceil$ and number of parameters $\mathcal{N} \leq M(2d + 404N \cdot (m+3) + 2N + 4) + 1$. By simply plugging-in $m = \lceil d^{\gamma} \rceil$, $N = \lceil d^{\alpha} \rceil$ and $M = 3$, we have $L = \mathcal{O}(d^{\gamma} \log_2(d))$ and $\mathcal{N} = \mathcal{O}\left(d^{\max\{\alpha+\gamma,1\}}\right)$.

*(II) $r = \mathcal{O}(1)$ and $M = \mathcal{O}(d^d)$.*

For the first term in (45), since $N = \lceil d^{\alpha} \rceil$, we know that $N^{-r} = \mathcal{O}(d^{-\alpha r})$.

For the second term in (45), since $N = \lceil d^{\alpha} \rceil$ with $0 < \alpha < 1$, we know that it is bounded by

$$\frac{\left(\frac{6}{\pi e}\right)^{\frac{d}{4}} d^{N + \frac{3d-4r-2}{8}} (2N+1)^{\frac{3d-4r}{4}}}{\sqrt{M}} \leq \frac{\left(\frac{6}{\pi e}\right)^{\frac{d}{4}} d^{d^{\alpha} + \frac{3d-4r+6}{8}} (3d^{\alpha})^{\frac{3d-4r}{4}}}{\sqrt{M}} \leq \frac{\left(\frac{6}{\pi e}\right)^{\frac{d}{4}} d^{d^{\alpha} + \frac{9}{8}d} 3^d}{\sqrt{M}}. \quad (47)$$

Take $M = \mathcal{O}(9^d d^{\frac{9}{4}d})$, multiply the bound (47) by $d^{d^\beta}$, take the logarithm, and find that for any $0 < \alpha, \beta < 1$,

$$\log\left(\left(\frac{6}{\pi e}\right)^{\frac{d}{4}} d^{d^\alpha + d^\beta + \mathcal{O}(1)}\right) \leq \frac{d}{4} \log\left(\frac{6}{\pi e}\right) + \left(d^\alpha + d^\beta + \mathcal{O}(1)\right) \log(d) \to -\infty,$$

as $d \to \infty$. Hence, there exists a constant $C_{\alpha,\beta} > 0$ depending only on $C_1$, $\alpha$, $\beta$ such that

$$\frac{\left(\frac{6}{\pi e}\right)^{\frac{d}{4}} d^{N + \frac{3d-4r-2}{8}} (2N+1)^{\frac{3d-4r}{4}}}{\sqrt{M}} \leq C_{\alpha,\beta} d^{-d^\beta} \leq C_{\alpha,\beta} d^{-\alpha r},$$

for $M = \mathcal{O}(9^d d^{\frac{9}{4}d})$. For the third term in (45), take $m = \lceil d^\gamma \rceil$ with $\max\{\alpha, \beta\} < \gamma < 1$, then there exists a constant $C_{\alpha,\beta,\gamma}$ depending on $\alpha, \beta, \gamma$ such that

$$d^{2N} \left(\log_2(2N)\right)^2 2^{-2m} < d^{2d^\alpha + 2} \left(2 + \log_2(d)\right)^2 2^{-2m} \leq d^{3d^\alpha} 2^{-2m} \leq C_{\alpha,\beta,\gamma} d^{-d^\beta} \leq C_{\alpha,\beta,\gamma} d^{-\alpha r},$$

where $\log_2(2d^\alpha + 2) \leq \log_2(4d^\alpha) < 2 + \log_2(d)$ is used in the first inequality, and the last inequality follows from the same argument as above, of multiplying with $d^{d^\beta}$ and taking the logarithm. Combining all the analyses above, we have

$$\left\|\tilde{f} - f\right\|_\infty \leq C'_{\eta,\alpha,\beta,\gamma} \|f\|_{W^r_\infty(\mathcal{S}^{d-1})} d^{-\alpha r},$$

where $C'_{\eta,\alpha,\beta,\gamma} > 0$ is a constant dependent on $\eta, \alpha, \beta, \gamma$, and $C_1$.

Recall from Proposition D.2.4, $\tilde{f}$ is a network with depth $L = (m+4)\lceil \log_2(2N) \rceil$ and number of parameters $\mathcal{N} \leq M(2d + 404N \cdot (m+3) + 2N + 4) + 1$. By simply plugging-in $m = \lceil d^\gamma \rceil$, $N = \lceil d^\alpha \rceil$ and $M = \mathcal{O}(9^d d^{\frac{9}{4}d})$, we have $L = \mathcal{O}(d^\gamma \log_2(d))$ and $\mathcal{N} = \mathcal{O}\left(9^d d^{\frac{13}{4}d}\right)$. $\qquad\square$

## F  PROOFS OF PROPOSITION 4.2, THEOREM 4.3 AND THEOREM 4.4

### F.1  PROOF OF PROPOSITION 4.2

**Proposition 4.2** *Set $\delta \in (0,1)$. Then, with probability at least $1 - \delta$, we have*

$$\mathcal{E}\big(\pi_B \widehat{f}_n\big) - \mathcal{E}(f_\rho) \leq C_{B,\delta,f} \cdot \left(\frac{Pdim(\mathcal{F}) \cdot \log(n)}{n} + \frac{\|f - f_\rho\|_\infty}{\sqrt{n}} + \|f - f_\rho\|_\infty^2\right), \qquad (48)$$

*where $C_{B,\delta,f}$ is an absolute constant dependent on $B, \delta, f$ independent on $n, r, d$.*

*Proof.* Since $\widehat{f}_n$ is an empirical risk minimizer in (12), we have $\mathcal{E}_D(\widehat{f}_n) \leq \mathcal{E}_D(f)$ for any fixed $f \in \mathcal{F}$ and $\mathcal{E}_D(\pi_B \widehat{f}_n) \leq \mathcal{E}_D(\widehat{f}_n)$. Then, we have a following decomposition:

$$\begin{aligned}
\mathcal{E}\big(\pi_B \widehat{f}_n\big) - \mathcal{E}(f_\rho) = & \left(\{\mathcal{E}(\pi_B \widehat{f}_n) - \mathcal{E}(f_\rho)\} - \{\mathcal{E}_D(\pi_B \widehat{f}_n) - \mathcal{E}_D(f_\rho)\}\right) \\
& + \left(\{\mathcal{E}_D(\pi_B \widehat{f}_n) - \mathcal{E}_D(f_\rho)\} - \{\mathcal{E}_D(f) - \mathcal{E}_D(f_\rho)\}\right) \\
& + \left(\{\mathcal{E}_D(f) - \mathcal{E}_D(f_\rho)\} - \{\mathcal{E}(f) - \mathcal{E}(f_\rho)\}\right) + \left(\mathcal{E}(f) - \mathcal{E}(f_\rho)\right) \\
\leq & \left(\{\mathcal{E}(\pi_B \widehat{f}_n) - \mathcal{E}(f_\rho)\} - \{\mathcal{E}_D(\pi_B \widehat{f}_n) - \mathcal{E}_D(f_\rho)\}\right) \qquad (49) \\
& + \left(\{\mathcal{E}_D(f) - \mathcal{E}_D(f_\rho)\} - \{\mathcal{E}(f) - \mathcal{E}(f_\rho)\}\right) + \left(\mathcal{E}(f) - \mathcal{E}(f_\rho)\right).
\end{aligned}$$

Let $\mathcal{F}_B := \{\pi_B f : \forall f \in \mathcal{F}\}$ and define two quantities:

$$\mathcal{S}_1(n, \mathcal{F}_B) := \{\mathcal{E}(f) - \mathcal{E}(f_\rho)\} - \{\mathcal{E}_D(f) - \mathcal{E}_D(f_\rho)\} \quad \forall f \in \mathcal{F}_B,$$
$$\mathcal{S}_2(n, \mathcal{F}) := \{\mathcal{E}_D(f) - \mathcal{E}_D(f_\rho)\} - \{\mathcal{E}(f) - \mathcal{E}(f_\rho)\} \quad \forall f \in \mathcal{F}.$$

**Step 1 : Control $\mathcal{S}_1(n, \mathcal{F}_B)$.** The following concentration inequality is needed for controlling the term.

**Lemma F.1.1** *[Theorem 11.4 of Györfi et al. (2002)] Assume $|y| \leq B$ almost surely and $B \geq 1$. Let $\alpha, \beta > 0$ and $0 < \varepsilon \leq 1/2$. If $\mathcal{F}'$ is a set of functions $f : \mathbb{R}^d \to [-B, B]$, then for any $f \in \mathcal{F}'$, we have*

$$\mathbb{P}\bigg( \mathcal{S}_1(n, \mathcal{F}') \leq \varepsilon(\alpha + \beta + \mathcal{E}(f) - \mathcal{E}(f_\rho)) \bigg)$$

$$\geq 1 - \sup_{\mathcal{D}} \mathcal{N}\bigg( \frac{\beta\varepsilon}{20B}, \mathcal{F}', \| \cdot \|_{L_1(D)} \bigg) \exp \bigg( -\frac{\varepsilon^2(1-\varepsilon)\alpha n}{214(1+\varepsilon)B^4} \bigg).$$

**Lemma F.1.2** *[Theorem 6 of Haussler (2018)] Let $B > 0$ and $\mathcal{F}'$ be a set of functions $f : \mathcal{X} \to [-B, B]$. Then for any $\varepsilon \in (0, B]$, there holds*

$$\mathcal{M}(\varepsilon, \mathcal{F}', \| \cdot \|_{L_1(D)})) \leq 2\bigg( \frac{2eB}{\varepsilon} \log \frac{2eB}{\varepsilon} \bigg)^{Pdim(\mathcal{F}')}. \tag{50}$$

Recall a classical relation between $\varepsilon$-packing number and $\varepsilon$-covering number that asserts

$$\mathcal{M}(2\varepsilon, \mathcal{F}, \| \cdot \|_{L_1(D)})) \leq \mathcal{N}(\varepsilon, \mathcal{F}, \| \cdot \|_{L_1(D)})) \leq \mathcal{M}(\varepsilon, \mathcal{F}, \| \cdot \|_{L_1(D)})), \tag{51}$$

for any $\varepsilon > 0$. Combining (50), (51), the facts $\log x < x$, $\forall x > 0$, and $\text{Pdim}(\mathcal{F}_B) \leq \text{Pdim}(\mathcal{F})$ (See Maiorov & Ratsaby (1999), page 297), we have the upper-bound on $\mathcal{N}(\varepsilon, \mathcal{F}_B, \| \cdot \|_{L_1(D)}))$ as follows:

$$\mathcal{N}(\varepsilon, \mathcal{F}_B, \| \cdot \|_{L_1(D)})) \leq 2\bigg( \frac{2eB}{\varepsilon} \log \frac{2eB}{\varepsilon} \bigg)^{\text{Pdim}(\mathcal{F}_B)} \leq 2\bigg( \frac{2eB}{\varepsilon} \bigg)^{2\text{Pdim}(\mathcal{F})}. \tag{52}$$

Then, taking $\varepsilon = \frac{1}{2}$, $\beta = \frac{1}{n}$ in Lemma F.1.1, using the upper-bound on covering number in (52) yields the lower bound for the confidence level in Lemma (F.1.1) as follows:

$$1 - \sup_{\mathcal{D}} \mathcal{N}\bigg( \frac{1}{40Bn}, \mathcal{F}_B, \| \cdot \|_{L_1(D)} \bigg) \exp \bigg( -\frac{\alpha n}{2568B^4} \bigg)$$

$$\geq 1 - C_B \cdot \exp \bigg( 2 \cdot \text{Pdim}(\mathcal{F}) \cdot \log(n) - \frac{\alpha n}{2568B^4} \bigg), \tag{53}$$

where $C_B > 0$ is some absolute constants dependent on $B$. Choosing $\alpha$ in (53) such that

$$\alpha = C_{B,\delta} \cdot \frac{\text{Pdim}(\mathcal{F}) \cdot \log(n)}{n}$$

with a properly chosen $C_{B,\delta} > 0$ absolute constant dependent on $B$ and $\delta$ yields the probability of following event is at least $1 - \frac{\delta}{2}$:

$$\mathcal{S}_1(n, \mathcal{F}_B) \leq \frac{1}{2}\bigg( C_{B,\delta} \cdot \frac{\text{Pdim}(\mathcal{F}) \cdot \log(n)}{n} + \frac{1}{n} + \mathcal{E}(\pi_B \widehat{f}_n) - \mathcal{E}(f_\rho) \bigg). \tag{54}$$

**Step 2 : Control $\mathcal{S}_2(n, \mathcal{F})$.** Define a random variable $\eta$ on $\mathcal{Z} = \mathcal{X} \times \mathcal{Y}$ to be

$$\eta(z) = (y - f(x))^2 - (y - f_\rho(x))^2.$$

Since $|\eta(z)| \leq (3B + \|f\|_\infty)^2$, then $|\eta(z) - \mathbb{E}[\eta(z)]| \leq 2(3B + \|f\|_\infty)^2$. It is also easy to see $\sigma^2 \leq \mathbb{E}[\eta^2] \leq (3B + \|f\|_\infty)^2 \|f - f_\rho\|_\infty^2$. Then, by the one-side Bernstein's inequality (see Lemma H.2), we have

$$\mathbb{P}\bigg( \mathcal{S}_2(n, \mathcal{F}) < \varepsilon \bigg) \geq 1 - \exp \bigg\{ -\frac{n\varepsilon^2}{2(3B + \|f\|_\infty)^2 (\|f - f_\rho\|_\infty^2 + \frac{2}{3}\varepsilon)} \bigg\}.$$

Taking $\frac{\delta}{2} = \exp \bigg\{ -\frac{n\varepsilon^2}{2(3B + \|f\|_\infty)^2 (\|f - f_\rho\|_\infty^2 + \frac{2}{3}\varepsilon)} \bigg\}$, $\mathcal{A} := 2(3B + \|f\|_\infty)^2$, $\mathcal{B} := \|f - f_\rho\|_\infty^2$ and solving the quadratic equation with respect to $\varepsilon$ yield the following inequalities with some absolute

constant $C_0'' > 0$ :

$$\varepsilon = \frac{\mathcal{A} \log\left(\frac{2}{\delta}\right) + \sqrt{\mathcal{A}^2 \log\left(\frac{2}{\delta}\right) + 9n\mathcal{A}\mathcal{B} \log\left(\frac{2}{\delta}\right)}}{3n}$$

$$\leq \frac{2\mathcal{A} \log\left(\frac{2}{\delta}\right)}{3n} + \sqrt{\mathcal{A}\mathcal{B} \cdot \frac{\log\left(\frac{2}{\delta}\right)}{n}}$$

$$\leq C_{B,f,\delta} \cdot \frac{\|f - f_\rho\|_\infty}{\sqrt{n}},$$

where in the first inequality, the facts $\sqrt{a+b} \leq \sqrt{a} + \sqrt{b}$ for $a, b > 0$ is used, and $C_{B,f,\delta}$ is a constant dependent on $C, B$ and $f$. Then, with probability at least $1 - \frac{\delta}{2}$, we have

$$\mathcal{S}_2(n, \mathcal{F}) \leq C_{B,f,\delta} \cdot \frac{\|f - f_\rho\|_\infty}{\sqrt{n}}. \tag{55}$$

**Step 3 : Combining Everything.** Note $\mathcal{E}(f) - \mathcal{E}(f_\rho) = \|f - f_\rho\|_{\rho_\mathcal{X}}^2 \leq \|f - f_\rho\|_\infty^2$. Then, plugging the (54) and (55) in (49) yields the claim. $\qquad\square$

### F.2 PROOF OF THEOREM 4.3

**Theorem 4.3** *Suppose $f_\rho \in W_\infty^r(\mathcal{S}^{d-1})$ with $r > 0$. A network $\widehat{f}_n$ from (6) with choices $N = \lceil n^{\frac{2}{3d+4r}}\rceil$, $M = \lceil n^{\frac{3d}{3d+4r}}\rceil$, and $m = \lceil \frac{r}{3d+4r} \log_2(n)\rceil$ yield the bound on the excess risk with probability at least $1 - \delta$ as follows:*

$$\mathcal{E}(\pi_M \widehat{f}_n) - \mathcal{E}(f_\rho)$$
$$\leq \mathcal{C}_{B,\eta,\delta,f} \cdot \max\left\{1, \frac{6rd}{(3d+4r)^2}(\log_2(n))^4, \left(\frac{6}{\pi e}\right)^{\frac{d}{2}} d^{2N + \frac{3d-4r-2}{4}}, d^{4N}\right\} \cdot n^{-\frac{2r}{2r+1.5d}}, \tag{56}$$

*where $\mathcal{C}_{B,\eta,\delta,f}$ depends on $B, \eta, \delta, f$ and independent on $d, r$ and $n$.*

*Proof.* Let $0 < \alpha < 1, m, N, M \in \mathbb{N}$ with $1 \leq N \leq d^\alpha + 1$. Then, for $f_\rho \in W_\infty^r(\mathcal{S}^{d-1})$, recall from Theorem D.2.4 that there exists a network

$$\tilde{f} \in \mathcal{F}(L, (d, 22NM, \ldots, 22NM, 1), \mathcal{N}) \tag{57}$$

with depth $L = (m+4)\lceil \log_2(2N)\rceil$ and number of parameters $\mathcal{N} \leq M(2d + 404N \cdot (m+3) + 2N + 4) + 1$ such that the corresponding network's approximation error is bounded as:

$$\left\|\tilde{f} - f_\rho\right\|_\infty \leq C_\eta'' \|f\|_{W_\infty^r(\mathcal{S}^{d-1})} \times$$
$$\max\left\{N^{-r}, \frac{\left(\frac{6}{\pi e}\right)^{\frac{d}{4}} d^{N + \frac{3d-4r-2}{8}}(2N+1)^{\frac{3d-4r}{4}}}{\sqrt{M}}, d^{2N}\left(\log_2(2N)\right)^2 2^{-2m}\right\}, \tag{58}$$

where $C_\eta''$ is a constant dependent on $\eta$, and independent on $d, r, N, M$ and $f$. Since the network width is $22NM$, the total number of units across the $L$-hidden layers (i.e., $\mathcal{U}$) of $\tilde{f}$ is bounded as

$$\mathcal{U} \leq 22NM \cdot (m+4)\lceil \log_2(2N)\rceil.$$

If $Nm = o(d)$, it is easy to see $\mathcal{N} \leq \mathcal{O}(Md)$. Recall from the result of Lemma H.1, the pseudo-dimension of function class $\mathcal{F}$ in (57) is bounded as follows: for some universal constants $C > 0$:

$$\text{Pdim}(\mathcal{F}) \leq C \cdot \left(mMd \cdot \lceil \log_2(N)\rceil \cdot \log\left(mMN\lceil \log_2(N)\rceil\right)\right). \tag{59}$$

Plug the (58), (59) in (48) from Proposition 4.2.

$$\mathcal{E}\big(\pi_M \widehat{f}_n\big) - \mathcal{E}\big(f_\rho\big) \le \mathcal{C}_{B,\eta,\delta,f} \times$$

$$\Bigg\{ \underbrace{\frac{mMd}{n} \log(n) \cdot \lceil \log_2(N) \rceil \cdot \log\big(mMN\lceil\log_2(N)\rceil\big)}_{\textbf{Bound for } \mathrm{Pdim}(\mathcal{F}) \cdot \log(n)/n}$$

$$+ \underbrace{\max\bigg\{ N^{-r}, \frac{\big(\frac{6}{\pi e}\big)^{\frac{d}{4}} d^{N+\frac{3d-4r-2}{8}}(2N+1)^{\frac{3d-4r}{4}}}{\sqrt{M}}, d^{2N}\big(\log_2(2N)\big)^2 2^{-2m}\bigg\}/\sqrt{n}}_{\textbf{Bound for } \big\|\tilde{f}-f_\rho\big\|_\infty/\sqrt{n}}$$

$$+ \underbrace{\max\bigg\{ N^{-2r}, \frac{\big(\frac{6}{\pi e}\big)^{\frac{d}{2}} d^{2N+\frac{3d-4r-2}{4}}(2N+1)^{\frac{3d-4r}{2}}}{M}, d^{4N}\big(\log_2(2N)\big)^4 2^{-4m}\bigg\}}_{\textbf{Bound for } \big\|\tilde{f}-f_\rho\big\|_\infty^2} \Bigg\},$$

$$\tag{60}$$

where $\mathcal{C}_{B,\eta,\delta,f}$ depends on $B$, $\eta$, $\delta$, $f$ and independent on $d, r$ and $n$. Then, under the regime $1 \le N \le d^\alpha + 1$ for some $0 < \alpha < 1$, choices of $m = \lceil \frac{r}{3d+4r}\log_2(n) \rceil$, $N = \lceil n^{\frac{2}{3d+4r}} \rceil$ and $M = \lceil n^{\frac{3d}{3d+4r}} \rceil$ make the fraction of the first term in (60) simple as follows:

$$\lceil \log_2(N) \rceil \cdot \log\big(mMN\lceil\log_2(N)\rceil\big) \le \frac{2}{3d+4r}\log_2(n)\log\bigg(\log_2(n) n^{\frac{3d+2}{3d+4r}}\frac{2r}{(3d+4r)^2}\lceil\log_2(n)\rceil\bigg)$$

$$\le \frac{6}{3d+4r}\big(\log_2(n)\big)^2.$$

Then, with the same choices of $m, N, M$ as above, we obtain the bound on the excess risk as :

$$\mathcal{E}\big(\pi_M \widehat{f}_n\big) - \mathcal{E}\big(f_\rho\big)$$
$$\le \mathcal{C}_{B,\eta,\delta,f} \cdot \max\bigg\{ 1, \frac{6rd}{(3d+4r)^2}(\log_2(n))^4, \big(\frac{6}{\pi e}\big)^{\frac{d}{2}} d^{2N+\frac{3d-4r-2}{4}}, d^{4N} \bigg\} \cdot n^{-\frac{4r}{4r+3d}}.$$

This concludes the proof. $\qquad\qquad\qquad\qquad\qquad\qquad\qquad\qquad\qquad\qquad\qquad\square$

## F.3  Proof of Theorem 4.4

**Theorem 4.4** *Suppose $f_\rho \in W_\infty^r([0,1]^d)$ with $r > 0$. A network $\widehat{f}_n$ from (9) with choices $N^H = \lceil n^{\frac{d}{2d+r}} \rceil$, and $m^H = \lceil \frac{d+r}{d+2r}\log_2(n) \rceil$ yield the bound on the excess risk with probability at least $1 - \delta$ as follows:*

$$\mathcal{E}\big(\pi_M \widehat{f}_n\big) - \mathcal{E}\big(f_\rho\big) \tag{61}$$
$$\le \mathcal{C}_{B,\eta,\delta,K} \cdot \max\bigg\{ \lceil\log_2(d+\lceil r\rceil)\rceil^2 (d+r)^d \cdot (\log_2(n))^3, \big(1+r^2+d^2\big)^2 6^{2d} + 3^{2r} \bigg\} \cdot n^{-\frac{2r}{2r+d}},$$

*where $\mathcal{C}_{B,\eta,\delta,K}$ depends on $B$, $\eta$, $\delta$, $K$ and independent on $d, r$ and $n$.*

*Proof.* From Theorem 5 of Schmidt-Hieber (2020), for any function $f_\rho \in \mathcal{C}_d^r([0,1]^d, K)$ and any integers $m \ge 1$ and $N \ge (r+1)^d \vee (K+1)e^d$, there exists a network

$$\tilde{f} \in \mathcal{F}\big(L, (d, 6(d+\lceil r\rceil)N, \ldots, 6(d+\lceil r\rceil)N, 1), \mathcal{N}, \infty\big) \tag{62}$$

with depth $L = 8 + (m+5)\big(1 + \lceil\log_2(d \vee r)\rceil\big)$ and the number of parameters $\mathcal{N} \le 141(1+d+r)^{3+d}N(m+6)$, such that

$$\big\|\tilde{f} - f_\rho\big\|_\infty \le (2K+1)(1+d^2+r^2)6^d N 2^{-m} + K3^r N^{-\frac{r}{d}}. \tag{63}$$

Then, similarly with the proof in Theorem 4.3, by the result of Lemma H.1, the pseudo-dimension of $\mathcal{F}$ in (62) can be bounded as

$$\text{Pdim}(\mathcal{F}) \leq C \cdot \left( m^2 N (d+r)^d \lceil \log_2(d \vee r) \rceil \log \left( (d + \lceil r \rceil) m N \lceil \log_2(d \vee r) \rceil \right) \right), \quad (64)$$

for some universal constants $C > 0$.

Plug the (63) and (64) in (48) from Proposition 4.2. Then, we obtain the bound on the excess risk as follows:

$$\mathcal{E}(\pi_M \widehat{f}_n) - \mathcal{E}(f_\rho) \leq \mathcal{C}_{B,\delta,K} \times$$
$$\left\{ \underbrace{\frac{m^2 N}{n} \log(n) \cdot (d+r)^d \lceil \log_2(d \vee r) \rceil \log \left( (d + \lceil r \rceil) m N \lceil \log_2(d \vee r) \rceil \right)}_{\textbf{Bound for } \text{Pdim}(\mathcal{F}) \cdot \log(n)/n}$$
$$+ \underbrace{\left( (1 + d^2 + r^2) 6^d N 2^{-m} + 3^r N^{-\frac{r}{d}} \right) / \sqrt{n}}_{\textbf{Bound for } \left\| \tilde{f} - f_\rho \right\|_\infty / \sqrt{n}}$$
$$+ \underbrace{\left( (1 + d^2 + r^2)^2 6^{2d} N^2 2^{-2m} + 3^{2r} N^{-\frac{2r}{d}} \right)}_{\textbf{Bound for } \left\| \tilde{f} - f_\rho \right\|_\infty^2} \right\}, \quad (65)$$

where $\mathcal{C}_{B,\delta,K}$ depends on $B$, $\delta$, $K$ and independent on $d, r$ and $n$. Note that we use $(a + b)^2 \leq 2a^2 + 2b^2$ for all $a, b \in \mathbb{R}$ for getting the bound on $\left\| \tilde{f} - f_\rho \right\|_\infty^2$. We choose the $N = \lceil n^{\frac{d}{2r+d}} \rceil$ and $m = \lceil \frac{d+r}{2r+d} \log_2(n) \rceil$. Then, a fraction of the first term in (65) can be bounded as:

$$\log \left( (d + \lceil r \rceil) \cdot \frac{d+r}{2r+d} \cdot \log_2(n) \cdot n^{\frac{d}{2r+d}} \cdot \log_2(d \vee r) \right) \leq \log_2 \left( (d + \lceil r \rceil)^2 n^2 \right).$$

Then, we obtain the bound on the excess risk as :

$$\mathcal{E}(\pi_M \widehat{f}_n) - \mathcal{E}(f_\rho)$$
$$\leq \mathcal{C}_{B,\eta,\delta,K} \cdot \max \left\{ \lceil \log_2((d + \lceil r \rceil) n^2) \rceil^2 (d+r)^d \cdot (\log_2(n))^3, \left( 1 + r^2 + d^2 \right)^2 6^{2d} + 3^{2r} \right\} \cdot n^{-\frac{2r}{2r+d}}.$$

This concludes the proof. $\qquad \square$

## G   PROOF OF APPROXIMATION RESULT FOR CNN: FANG ET AL. (2020)

**Theorem C.1.** *Let* $2 \leq S \leq d$, $0 < \alpha < 1$, *and* $B, N, M \in \mathbb{N}$ *with* $1 \leq N \leq d^\alpha + 1$. *Let* $J \geq \lceil \frac{Md-1}{S-1} \rceil$, $\mathcal{D}_1 = (2B+3) \lfloor (d + JS)/d \rfloor$, *and* $\mathcal{D}_2 = \lfloor (d + JS)/d \rfloor$. *Then, for any function* $f \in W_\infty^r(\mathcal{S}^{d-1})$ *with* $r > 0$, *there exists a network* $\tilde{f}^{CNN} \in \mathcal{H}_{J,\mathcal{D}_1,\mathcal{D}_2,S}$ *with the number of parameters* $\mathcal{N} \leq J(3S + 2) + M + 2B + 4$ *such that*

$$\left\| f - \tilde{f}^{CNN} \right\|_\infty \leq C_{\eta,\alpha}'' \|f\|_{W_\infty^r(\mathcal{S}^{d-1})} \max \left\{ N^{-r}, \frac{\left( \frac{6}{\pi e} \right)^{\frac{d}{4}} d^{N + \frac{3d-4r-2}{8}} (2N + 1)^{\frac{3d-4r}{4}}}{\sqrt{M}}, d^{2N} \frac{r}{r-1} \frac{N^2}{B} \right\}, \quad (66)$$

*where* $C_{\eta,\alpha}''$ *is a constant dependent on* $\eta, \alpha$, *and independent on* $d, r, N, M$ *or* $f$.

*Proof.* By the inequality (5.9) in the paper Fang et al. (2020), we have

$$\left\| \widehat{L}_{N,M}^{\boldsymbol{y}}(f) - \tilde{f}^{CNN} \right\|_\infty \leq \|f\|_{W_\infty^r(\mathcal{S}^{d-1})} \|\xi_{N,r} - \mathcal{L}_t(\xi_{N,r})\|_{C[-1,1]}, \quad (67)$$

where $\mathcal{L}_t(\xi_{N,r})(t)$ is an univariate function for $t \in [-1, 1]$ defined in Lemma 7 in Fang et al. (2020). Now, we work on controlling $\|\xi_{N,r} - \mathcal{L}_t(\xi_{N,r})\|_{C[-1,1]}$.

$$
\begin{aligned}
\|\xi_{N,r} - \mathcal{L}_t(\xi_{N,r})\|_{C[-1,1]} &\leq 2\omega(\xi_{N,r}, 1/B) \\
&\leq \frac{2}{B}\|\xi'_{N,r}\|_{C[-1,1]} \\
&\leq \frac{8N^2}{B}\|\xi_{N,r}\|_{C[-1,1]} \\
&\leq \frac{8N^2}{B}\sum_{k=1}^{2N} k^{-r}\mathcal{N}(k, d) \\
&\leq \frac{8C_\alpha N^2}{B}d^{2N}\sum_{k=1}^{2N} k^{-r} \\
&\leq \frac{8C_\alpha N^2}{B}d^{2N}\frac{r}{r-1}.
\end{aligned}
$$

In the first inequality, we use Lemma 7 in Fang et al. (2020), where $\omega(\xi_{N,r}, 1/B)$ is a modulus continuity of $\xi_{N,r}$ given by

$$
\omega(\xi_{N,r}, 1/B) = \sup_{|t| \leq 1/B} \{|\xi_{N,r}(\nu) - \xi_{N,r}(\nu + t)| : \nu, \nu + t \in [-1, 1]\}.
$$

In the second inequality, we use the definition of modulus continuity of $\xi_{N,r}$. In the third inequality, since $\xi_{N,r}$ is an algebraic polynomial of degree at most $2N$, by Markov's inequality, we have $\|\xi'_{N,r}\|_{C[-1,1]} \leq (2N)^2\|\xi_{N,r}\|_{C[-1,1]}$. In the fourth inequality, we have the bound $\|\xi_{N,r}\|_{C[-1,1]} \leq \sum_{k=1}^{2N} k^{-r}\mathcal{N}(k, d)$ by Corollary 1.2.7 of Dai & Xu (2013). Employing Stirling's formula $d! = \sqrt{2\pi d}\left(\frac{d}{e}\right)^d\left(1 + \mathcal{O}(1/d)\right)$ in $\mathcal{N}(k, d)$ yields $\mathcal{N}(k, d) \leq C_\alpha d^k$ for $0 \leq k \leq 2N$ and some constant $C_\alpha$ depending on $\alpha$ but independent of $d$. In the last inequality, we used the following inequality:

$$
\sum_{k=1}^{2N} k^{-r} \leq 1 + \sum_{k=2}^{2N}\int_{k-1}^{k} x^{-r}dx = 1 + \underbrace{\int_{1}^{2N} x^{-r}dx}_{\leq \frac{1}{r-1}} \leq \frac{r}{r-1}.
$$

Now, we combine our result from Lemma $D.1.2$ and Lemma $D.1.4$, and conclude the bound in (66). □

# H USEFUL LEMMAS

**Lemma H.1** *[Theorem 6 of Bartlett et al. (2019)] Consider the function class $\mathcal{F}$ computed by a feed-forward neural network architecture with $\mathcal{N}$ parameters and $\mathcal{U}$ computation units arranged across $L$ layers. Suppose that all non-output units have piecewise-polynomial activation functions with $p + 1$ pieces and degrees no more than $d$, and the output unit has the identity function as its activation function. Then the VC-dimension and pseudo-dimension of class $\mathcal{F}$ is upper bounded by*

$$
VCdim(\mathcal{F}), Pdim(\mathcal{F}) \leq C \cdot \left(L\mathcal{N}\log(p \cdot \mathcal{U}) + L^2\mathcal{N}\log(d)\right),
$$

*with some universal constants $C > 0$.*

**Lemma H.2** *[Theorem 2.8.4 of Vershynin (2018)] Let $\eta$ be a random variable on a probability space $\mathcal{Z}$ with mean $\mathbb{E}(\eta) = \mu$, variance $\sigma^2(\eta) = \sigma^2$, and satisfying $|\eta(z) - \mathbb{E}(\eta)| \leq B_\eta$ for almost $z \in \mathcal{Z}$. Then, for any $\varepsilon > 0$,*

$$
\mathbb{P}\left\{\frac{1}{n}\sum_{i=1}^{n}\eta(z_i) - \mu < \varepsilon\right\} \geq 1 - \exp\left\{-\frac{n\varepsilon^2}{2\left(\sigma^2 + \frac{1}{3}B_\eta\varepsilon\right)}\right\}.
$$

