# OpenReview forum: "Approximation and non-parametric estimation of functions over high-dimensional spheres via deep ReLU networks"
_ICLR.cc/2023/Conference — ICLR 2023 poster_

### Official Review · Reviewer_52e1 · 2022-10-24

**Confidence:** 4
**Correctness:** 4
**Technical Novelty And Significance:** 2
**Empirical Novelty And Significance:** Not applicable
**Recommendation:** 8

**Clarity, Quality, Novelty And Reproducibility:**

Clarity: The presentation is clear.
Quality: Fair.
Novelty: Fair.
Reproducibility: N.A.

**Strength And Weaknesses:**

Strength:
1.In general, the paper is well organized with a clear presentation.
2.The paper derived new approximate results for Sobolev functions defined on spheres in the sense of tracking the explicit dependency on $d$ in the constant factor compared to existing ones.
3.The paper talked about the results when the smoothness $r$ varies with respect to dimension $d$.

Weakness
1.The obtained estimation error rate does not achieve the expected optimal minimax rate.  Regarding to the comparison in Table 1, it looks a bit weird for me since sphere $\mathcal{S}^{d-1}$ is a subset of the cube $[0,1]^d$, but the estimation rate on a subset
Is even slower. Is there a way to fix it? Or is there a reasonable explanation?

2.Regarding to the main result in Theorem 3.1, the approximation error has basically dependence $d^{N+d} \times N^{d}/\sqrt{M}$. This result seems also cursed by $d$ in numerator, otherwise one has to set a very large $M$, e.g. $M=d^d$ to get a good approximation accuracy. Please explain more on this.

3.The condition $r=O(d)$ is restrictive. The author may explain more on this condition, especially it’s better to give an example on in what scenarios the condition $r=O(d)$ holds. In addition, I wonder if the condition $r=O(d)$ can be plugged in other existing results, e.g., Schmidt-Hieber (2020) to have a comparison with the obtained results in these $d, r$ varying cases. If it’s not possible, please also explain why $r=O(d)$ can not directly be plugged in other results.

4.The comparison can be fairer since Schmidt-Hieber (2020) focus on the approximation on hyper-cubes instead of sphere. The authors can compare their results with Fang et al. (2020) and Feng et al. (2021) since a large part of the proof follows that of Fang et al. (2020). The most notable difference between Fang et al. (2020) and this paper is that this paper track the explicit dependence of the constant factor on $d$. In addition, though Fang et al. (2020) and Feng et al. (2021) considered CNN approximations, Zhou, D.X. (2020). has shown that any CNN can be equivalently computed by a FNN with parameters at most 8 times larger than that of CNN. In light of this, the author should compare their results with Fang et al. (2020) and Feng et al. (2021).

5.Within the studies of ReLU FNN approximation, the author can also compare with more recent results which have already improved the results in Schmidt-Hieber (2020). For example, Shen, Yang, and Zhang. (2020) and Lu, Shen, Yang, and Zhang. (2021) have already demonstrated the clear dependence on dimension $d$ for ReLU approximation results. In line with these works, Jiao, Shen, Lin and Huang (2021) has also shown the explicit dependence of ReLU approximation results on $d$ as well as obtained polynomial dependence results.

Minor comment:
1.For reference, the issues about the dependence on dimension $d$ was also discussed in Ghorbani et al. (2020).
2.The obtained approximation results are for Sobolev function space $W^r_\infty$, which is actually Hölder function space. This is also mentioned in Appendix A.

Reference

SCHMIDT-HIEBER, J. (2020). Nonparametric regression using deep neural networks with ReLU activation function(with discussion). Ann. Statist. 48 1875–1897.

Ding-Xuan Zhou. Theory of deep convolutional neural networks: Downsampling. Neural Networks,
124:319–327, 2020.

SHEN, Z., YANG, H. and ZHANG, S. (2020). Deep network approximation characterized by number of neurons.
Commun. Comput. Phys. 28 1768–1811.

LU, J., SHEN, Z., YANG, H. and ZHANG, S. (2021). Deep network approximation for smooth functions. SIAM
Journal on Mathematical Analysis 53 5465–5506.

Jiao, Y., Shen, G., Lin, Y., & Huang, J. (2021). Deep nonparametric regression on approximately low-dimensional manifolds. arXiv preprint arXiv:2104.06708.

GHORBANI, B., MEI, S., MISIAKIEWICZ, T. and MONTANARI, A. (2020). Discussion of: “Nonparametric regression using deep neural networks with ReLU activation function”. Ann. Statist. 48 1898–1901.

**Summary Of The Paper:**

This paper concerns the approximation power of ReLU FNNs for Sobolev (or actually Hölder) smooth functions defined on spheres. This paper constructs ReLU FNNs with certain architectures to obtain their approximation results. The main idea is to leverage approximation capability of the homogeneous harmonic polynomials on spheres. The obtained results are mainly compared with those in Schmidt-Hieber (2020), which nevertheless concerns the ReLU FNN approximation on functions defined on $d$-cubes. By comparison, it is claimed that their approximation results can alleviate the dependence of network size (number of total parameters) on dimension $d$ to achieve similar approximation error bounds in Schmidt-Hieber (2020). In addition, the paper has also discussed on the approximation results when the smoothness of the target function $r$ varies w.r.t $d$.

**Summary Of The Review:**

In general, the paper is well organized with a clear presentation. This paper concerns the approximation power of ReLU FNNs for Hölder smooth functions defined on spheres. This paper constructs ReLU FNNs with certain architectures to obtain their approximation results, while a large part of the proof follows that of Fang et al. (2020). The most notable difference between Fang et al. (2020) and this paper is that this paper track the explicit dependence of the constant factor on $d$. Besides, several major questions raised in “Weakness” part need to be addressed.

---

> ### Author Response · Authors · 2022-11-11
> **Response to reviewer 3's comments**
>
> Thank you so much your constructive comments!
> If there still are some concerns, please let us know! We would love to have the interactions with you!
> For conciseness, we will not repeat your comments, and will give the answers directly.
> ---------------------------------------------------------------------------------------------------------------------------------------------------------
> 1.
> This is a great point.
> Thanks for pointing out.
> In light of Whitney's extension theorem, it is obvious that the convergence rate of excess risk for $f \in W^{r}(S^{d-1})$  should achieve the minimax optimal rate $n^{-\frac{2r}{2r+d}}$ same as for estimating $f \in W^{r}_{\infty}(\mathcal{C}^{d})$.
> However, from this perspective, it is not possible to track the explicit dependence on $d$ in prefactor of the rate.
>
> From Proposition $4.2$, we track this dependence with the combination of our own approximation result.
> First of all, in order to achieve the minimax learning rate in $n$ (e.g., sample size); that is, $n^{-\frac{2r}{2r+d}}$, we need to achieve the optimal approximation rate, known as $\mathcal{O}(\mathcal{N}^{-\frac{r}{d}})$, where $\mathcal{N}$ denotes the number of active parameters in the network.
> This can be easily checked in Schmidt-Hieber's result.
>
> But this cannot be achieved in our Theorem 3.1, the result of approximation theorem. The main reason arises from the employment of Sobolev embedding theorem (proposition D.1.3) and of concentration inequality from Smale and Zhou (2007) (Lemma E.2.1). In this process, there are some extra factors multiplied leading the rate sub-optimal, reflected in the term $(2N+1)^{\frac{3d-4r}{4}}$.
> It would be an interesting research direction, if we can develop a nice mathematical framework eliminating this extra factor!
>
> This is further elaborated in Appendix B in our revised manuscript.
> ---------------------------------------------------------------------------------------------------------------------------------------------------------
> 2.
> This is correct!
> The exact dependence is $d^{N+\frac{3d-4r-2}{8}}N^{\frac{3d-4r}{4}}/\sqrt{M}$.
> As you mentioned, when data dimension $d$ is large and $r=\mathcal{O}(1)$, large $M$ (e.g., $M=d^{d}$) is required.
> However, this effect is ameliorated when the order of $r$ increases, and when its order is $\mathcal{O}(d)$, $M$ can be any arbitrary integer to get a good approximation rate, avoiding the
> ``curse of dimensionality''.
> This is the point where an interaction with $r$ and $d$ comes from.
> To the best of our knowledge, this is not something observed in the current literature on approximation theory of neural networks.
> The result in Corollary $3.3.$ is derived from the above observation.
> We mentioned about this observation in the original draft (Page $6$, third paragraph), but we admit some sentences can be confusing.
> We hope that the modified sentences (colored in red) are more clear to the reviewer and to future readers of this paper.
> ---------------------------------------------------------------------------------------------------------------------------------------------------------
> 3.
> Thanks for the comment!
> Admittedly, the condition $r=\mathcal{O}(d)$ is restrictive in a sense that the function space of interests becomes small.
> However, it contains some interesting examples.
> Notably functions from reproducing kernel Hilbert spaces (RKHS) generated by $C^{\infty}$ kernels such as Gaussian kernels defined on sphere
> belong to the space $W^{r}_{\infty}(\mathcal{S}^{d-1})$ with $r=\mathcal{O}(d)$.
> We include this in the revised manuscript.
>
> Regarding the second question in this comment, to the best of our knowledge, and after the careful readings of Schmidt-Hieber's proof, we noticed no restrictions both on the smoothness $r$ (in his paper $\beta$) and on the data dimension $d$ are placed, and the statement holds for any $r > 0$ and $d\in\mathbb{N}_{+}$, and the relation $r=\mathcal{O}(d)$ can be plugged in his result.

---

> > ### Author Response · Authors · 2022-11-11
> > **Response to reviewer 3's comment.**
> >
> > 4.Indeed, our work is heavily influenced by Fang et al. (2020).  But as you pointed out we track the explicit dependence in the pre-factor $d$ (Lemma D.1.4), and additionally we constructed the neural network with deep ReLU FNN structure that can approximate $\widehat{L}_{N,M}^{\boldsymbol{y}}(f)$.
> >
> > As you suggested, we consider the case when the approximator is deep ReLU CNN followed by downsampling operations and very few fully-connected layers. This is exactly same architecture considered in the paper Fang et al. (2020), and by applying our result (Lemma B.1.4), we get a theorem stated in Appendix C.1. in our revised manuscript!
> >
> > The only part we need to pay attention to is bounding the term $\| \widehat{L}_{N,M}^{\boldsymbol{y}}(f) - {f}^{\text{CNN}} \|$, and track the explicit dependence on $d$ in the bound, and its proof is deferred in Appendix G in the revised manuscript.
> >
> > We consider the case \textit{\textbf{\underline{$r=\mathcal{O}(d)$ and any integer $M\geq 1$}}}.
> > In this case, our result for deep ReLU FNN shows that $\mathcal{O}(d^{-d^{\beta}})$ can be achieved for the approximation, with at most $\mathcal{O}(d^{2})$ active parameters.
> > In light of the result from Zhou, D.X. (2020), we also should expect the same results for CNN with downsampling operation.
> >
> > In order to get an approximation rate $\mathcal{O}(d^{-d^{\beta}})$ for some $0<\beta<1$,
> > controlling the first two term in (1.1) is same with that of our proof in the appendix C.6.
> > We only need to pay attention to the last term.
> > Since $1\leq N \leq d^{\alpha}+1$, for some $0\leq\alpha<1$, we have
> > $$
> >     \frac{r}{r-1} \cdot d^{2N} N^{2}/B
> >     \leq 8 \cdot d^{2d^{\alpha}+2}d^{2\alpha}/B
> >     \leq 8 \cdot d^{2d^{\alpha}+4}/B
> >     \leq C \cdot d^{-d^{\beta}},
> > $$
> > for some constant $C>0$ independent with $d>0$ and $r>0$.
> > The rate $d^{-d^{\beta}}$ is obtainable only when $B=\mathcal{O}(d^{d})$.
> > Then, the number of parameters $\mathcal{N} \leq J(3S+2) + M + 2B + 4$ is bounded by $\mathcal{O}(d^{d})$.
> > This is an unsatisfatory result.
> > However, we firmly believe this result can be improved, and leave it as an open question for the future work.
> > -----------------------------------------------------------------------------------------------------------------------------------------------------------------------------
> > 5.
> > Thanks for the papers!
> > We do enjoy reading the papers.
> > Please refer the Appendix C.2. in our revised manuscript for the comparisons!
> > Additionally, we create another Table 2, please also refer to that Table.
> > -----------------------------------------------------------------------------------------------------------------------------------------------------------------------------
> > Minor comments : Thanks for the comment!
> > We mentioned Ghorbani et al. (2020) in the Appendix C.2. in our revised manuscript.
> > For the second remark, it is correct. We add the additional statement on this fact, right below the equation (5) in the revised manuscript for the clarification.

---

> > ### Comment · Reviewer_52e1 · 2022-11-13
> > **Thanks for your reply.**
> >
> > Thanks for the authors' reply, which removes my concerns.  I think this work will do a nice contribution to this conference.  After carefully read all the responses to all the reviewers, I increase my score.

---

### Official Review · Reviewer_JPQC · 2022-10-24

**Confidence:** 2
**Correctness:** 4
**Technical Novelty And Significance:** 3
**Empirical Novelty And Significance:** 3
**Recommendation:** 6

**Clarity, Quality, Novelty And Reproducibility:**

Clarity: This paper is clearly written and well organized. I find it easy to follow.

Quality: This paper is technically sound.

Novelty: I think the results in this paper are significant, as explained above.

**Strength And Weaknesses:**

The main results in this paper will certainly help us have a better understating of the approximation of deep neural networks from a theoretical way. The techniques in this paper may be extended to more general architectures.

**Summary Of The Paper:**

The authors study the approximation of functions using neural networks with ReLU activations. They showed that, for the functions defined on some Sobolev spaces over the d-dimensional unit, there exists a fast approximation error rate of $d^{-d^\beta}$ by some neural networks, which is very sharp. Also, the construction of the suggested neural networks is given.

**Summary Of The Review:**

In summary, I think the contribution of this paper is enough and this paper is suitable for ICLR, as explained above.

---

> ### Author Response · Authors · 2022-11-11
> **Response to Reviewer 2**
>
> Thank you so much your kind words and comments! If you have any further questions, please let us know!

---

### Official Review · Reviewer_JVzW · 2022-10-24

**Confidence:** 5
**Correctness:** 4
**Technical Novelty And Significance:** 4
**Empirical Novelty And Significance:** Not applicable
**Recommendation:** 8

**Clarity, Quality, Novelty And Reproducibility:**

The paper is well-written with a sufficient literature review. The theoretical analysis is neat and well-presented. The results are novel, significant, and of great interest.

**Strength And Weaknesses:**

$\textbf{Strength}$

1. To my best knowledge this is a novel contribution that broadens the understanding of deep ReLU FNN for approximating Sobolev functions defined on the high-dimensional sphere ($d\rightarrow\infty$), a topic of great interest to the ICLR community

2. The paper explicitly describes the dependence of the approximation error and excess risk on the data dimension $d$, which shows that the curse of dimensionality may be avoided when $r=O(d)$.

3. Excess risk bounds are derived through the lens of pseudo-dimension, which removes the boundedness requirement of the weight parameters in the literature, e.g., Schmidt-Hieber (2020).

4. A different phenomenon is observed compared to the case for $d$-dimensional cubes, which is interesting and novel.

$\textbf{Concerns}$

1. Can the results or analysis be extended to other neural networks, e.g. convolutional neural networks?

2. Will it make any difference if we consider the regime $r=O(d^\alpha)$ for some $0<\alpha<1$?

Minor typos:

1. In Table 1, 4th row & 3rd column, should $O(d^2)$ be $O(d^d)$?

**Summary Of The Paper:**

The paper investigates the theoretical properties of deep feed-forward neural networks (FNN) with ReLU activation function when the target function belongs to a Sobolev space defined over the $d$-dimensional unit sphere with smoothness index $r>0$. The bounds for both approximation error and excess risk are derived for two different regimes as $d\rightarrow\infty$, which explicitly depend on the data dimension $d$ as follows

1. In the regime where $r=O(1)$: at most $d^d$ active parameters to get the approximation rate of $d^{-C}$ with $C>0$ being a constant; Moreover, the excess risk bound has a $d^d$ factor

2. In the regime where $r=O(d)$: at most $d^2$ active parameters to get the approximation rate of $d^{-d^{\beta}}$ with $0<\beta<1$; Moreover, the excess risk bound has a $d^{O(1)}$ factor

The authors make comparisons to the case where the input space is d-dimensional cubes and highlight the novelty of their results.

**Summary Of The Review:**

The paper presents novel results on how the approximation rate (excess risk) of deep ReLU FNN depends on the data dimension when the input space is a high-dimensional sphere, which provides new insights into the approximation analysis of deep neural networks and will be of great interest to the community of deep learning theory. The intuition is clearly explained and the theorems are technically sound.

---

> ### Author Response · Authors · 2022-11-11
> **Response to reviewer 1**
>
> Thank you so much your kind words and comments!
> We have answers to each of your comments. If you have further question, please let us know!
>
> 1. Can the results or analysis be extended to other neural networks, e.g. convolutional neural networks?
>
> Ans : This is an important question, and the answer to this question is yes, but we need further research.
> Based on your comment and Reviewer 3's comment, we applied our own result (Lemma D.1.4) to CNN approximator.
> We use the exactly same architecture suggested by Fang et al. (2020) : a deep ReLU CNN followed by downsampling operations and very few fully-connected layers.
> Based on the architecture suggested in Fang et al. (2020), we further tracked the dependence on $d$ in the pre-factor for approximating $\widehat{L}_{N,M}^{\boldsymbol{y}}(f)$ with the suggested network.
>
> This is stated in the Appendix C.1 in our revised manuscript, and our conclusion is:
> even when $r=\mathcal{O}(d)$, the suggested network by Fang et al. (2020), cannot avoid the curse, since it requires at most $\mathcal{O}(d^{d})$ active parameters.
> However, this is something in contrary to what is expected based on the existing literature.
> In light of the result from Zhou, D.X. (2020) (Theorem 2), we also should expect the suggested network should avoid the curse when $r=\mathcal{O}(d)$.
> Please, refer the further remarks by us to the question raised by Reviewer 3!
>
> 2. Will it make any difference if we consider the regime $r=\mathcal{O}(d^{\alpha})$ for some $0<\alpha<1$ ?
>
> Ans : Thanks for the interesting question.
> If we consider the regime $r=\mathcal{O}(d^{\kappa})$ for some $0 < \kappa < 1$, then, by following the same proof of Corollary 3.3. in our paper,
> we will get the approximation rate up to $\mathcal{O}(d^{-d^{\kappa}})$.
> In order to achieve the $\mathcal{O}(d^{-d^{\kappa}})$ error rate, the required order of $M$ should be roughly $d^{d-d^\kappa}$.
> As $\kappa$ gets closer to $1$, then this achieves our original result, when $r=\mathcal{O}(d)$.
> But, please note that it will have little effect on the depth of our constructed network.
>
> Minor typos:
> In Table 1, 4th row & 3rd column, should  $\mathcal{O}(d^{2})$ be $\mathcal{O}(d^{d})$?
>
> Ans : No, this is not a typo. However, we changed it to the expression $\mathcal{O}(nd)$.
> The active parameter $\mathcal{N}$ is essentially bounded by $\mathcal{O}(Md)$ regardless of the order of $r$ by Theorem $3.1$.
> Based on this, we choose $M=n^{\frac{3d}{3d+4r}}$ in Theorem 4.3, and this yields the bound $\mathcal{O}(nd)$.
> The original order $\mathcal{O}(d^{2})$ came from the assumption $n \ll d$, but we found this assumption cannot be applied in the cases of $r=\mathcal{O}(1)$ in Theorem 4.3. and Theorem 4.4.
> This is because the rate for the excess risk in these two cases cannot go to zero, under this assumption.
> For these two cases, you need $n\succsim d^{d}$ sample sizes for good convergence rate.
> So in our revised manuscript, we eliminate this assumption.
> Thanks for pointing out!

---

> > ### Comment · Reviewer_JVzW · 2022-11-18
> > **Response to authors**
> >
> > I would like to thank the authors for their responses and clarifications. I maintain my positive assessment of this work.

---

### Author Response · Authors · 2022-11-11
**Thanks for your comments! reviewers!**

We do appreciate for all of your constructive and positive comments on our paper.
Hope that our revised manuscript makes sense to you.
Our revised manuscript includes following additional insights.

1. We add an additional technical insight on why we have sub-optimal rate in excess risk for estimating $f \in W^{r}(S^{d-1})$. (Appendix B)

2. We apply our own result to CNN (Fang, et al. 2020), and conclude we need further research in light of D-X. Zhou, since in the current result, the suggested CNN in Fang et al. 2020 requires $\mathcal{N}=\mathcal{O}(d^{d})$ even in $r=\mathcal{O}(d)$. (Appendix C.1.)

3. We compare our own result with the recent results on approximation theory of deep ReLU FNN. (Appendix C.2.)

We feel your constructive comments improve our paper. Appreciate again!

Sincerely,
Authors

---

### Decision · Program_Chairs · 2023-01-20

**Decision:**

Accept: poster

**Justification For Why Not Higher Score:**

The paper makes a solid contribution to the understanding of approximation properties of ReLU networks, and identifying an intriguing phenomenon. The broad approach follows an existing line of argument (through harmonic polynomials), but clarifies the dependence on dimension.

**Justification For Why Not Lower Score:**

The paper advances our understanding of approximation properties of neural networks, clarifying dimension dependencies (which are important for high-dimensional problems), identifying highly smooth regimes where high-dimensional learning is possible, and identifying intriguing contrasts with learning over the cube.

**Metareview: Summary, Strengths And Weaknesses:**

The paper studies the approximation of Sobolev functions over the high-dimensional spheres, using deep fully connected neural networks. It develops bounds on the width and depth required for accurate approximation, and on the number of active parameters. These bounds have three interesting features compared to previous work. First, they explicitly track all dependencies on the dimension d. Second, it shows that for extremely smooth functions (Sobolev parameter r ~ d), we can have rates which are polynomial in dimension d. Third, it contrasts these rates with analogous results for the Sobolev functions over the cube (including upper and lower bounds) and shows that this phenomenon does not arise over the cube. Proofs go through an expansion of the target function in spherical harmonics, which can then be approximated individually by subnetworks. The paper upper bounds the risk associated with the best approximating network; the approach differs from corresponding results on the cube, in that it does not require the weights of the approximating network to be bounded.

Reviewers found that the paper makes a solid contribution to the understanding of the approximation capabilities of ReLU networks, in particular clarifying dimension dependencies (which are critical for high-dimensional data) and illustrating an interesting contrast with learning over the cube.

**Note From Pc:**

if the above contains the word "oral" or "spotlight" please see: "oral" presentation means -> notable-top-5% and "spotlight" means -> notable-top-25%. As stated in our emails, we are disassociating presentation type from AC recommendations